 SciPost Phys. Lect. Notes 45 (2022)

# An introduction to axions and their detection

**Igor G. Irastorza**

Center for Astroparticle and High Enery Physics (CAPA),
Universidad de Zaragoza, 50009 Zaragoza, Spain.

Igor.Irastorza@unizar.es

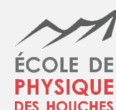

*Part of the Dark Matter*
*Session 118 of the Les Houches School, July 2021*
*published in the Les Houches Lecture Notes Series*

## Abstract

In these notes I try to introduce the reader to the topic of axions: their theoretical motivation and expected phenomenology, their role in astrophysics and as a dark matter candidate, and the experimental techniques to detect them. Special emphasis is made in this last point, for which a relatively updated review of worldwide efforts and future prospects is made. The material is intended as an introduction to the topic, and it was prepared as lecture notes for Les Houches summer school 2021. Abundant references are included to direct the reader to deeper insight on the different aspects of axion physics.

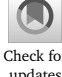

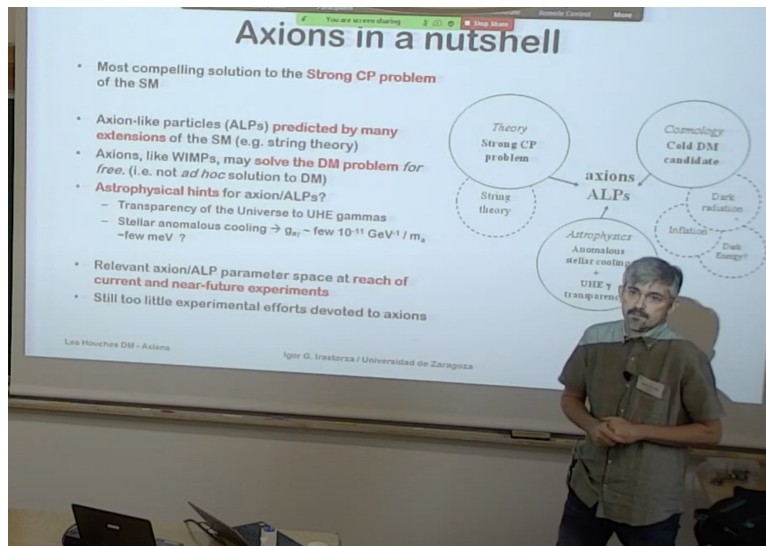

# 1 Introduction

Axion-like particles (ALPs) appear in many extensions of the Standard Model (SM), typically those with the spontaneous breaking of one or more global symmetries at high energies. ALP models are invoked in attempts to solve shortcomings of the SM, but also of cosmological or astrophysical unexplained observations. Most relevantly, ALPs are ideal dark matter candidates. In addition, and not exhaustively, ALPs have been invoked to solve issues as diverse as the hierarchy problem in the SM, the baryon asymmetry of the Universe, inflation, dark energy, dark radiation, or to explain the anomalous cooling observed in several types of star. The QCD axion is the prototype particle of this category, proposed long ago to solve the strong-CP problem of the SM. Still the most compelling solution to this problem, it remains maybe the strongest theoretical motivation for the "pseudoscalar portal" to new physics.

Typical axion models are constrained to very small masses below ∼1 eV. Because of that, signatures of these particles are not expected at accelerators, and novel specific detection techniques are needed[1]. The particular combination of know-hows needed for these experiments, some of them not present in typical high-energy physics (HEP) groups (and including, among others, high-field magnets, super-conduction, radiofrequency (RF) techniques, X-ray optics & astronomy, low background detection, low radioactivity techniques, quantum sensors, atomic physics, etc...), and their effective interplay with axion particle physicists is an important challenge in itself. We will focus here on the detection efforts of these low-energy axions[2].

These notes have been written in support of a course given in Les Houches summer school 2021. As such they are intended as an introduction to the subject of axion physics. The emphasis is put in the experimental efforts to search for these particles, although an introduction to the theory and main phenomenology, both in cosmology and astrophysics, are also included. The students seeking a more in-depth treatment of some of the topics presented can consult the many references included throughout the text. A good point to start are the modern reviews [2] and [3]. Both are recent efforts to describe a rapidly evolving subfield. Much of the material presented here is based on them. The latter is a thorough review of the theory and the latest phenomenological developments on axions, while the former has an emphasis on detection and experiments. Another interesting reference [4] includes axions in the more generic portfolio of searches for dark matter, recently compiled as a strategy document for the APPEC committee. And yet another reference is [5], this one not linked to the dark matter issue, in which axions and ALPs (the pseudoscalar "portal") are presented in the wider context of possible extensions of the SM including "feebly interacting particles", and thus encompassing also other "portals" for new physics including new fermion (e.g. neutrino-like), vector (like in some light dark matter models) or scalar (e.g. Higgs-like) particles. Finally, let us also mention a very recent textbook [6] which includes pedagogical material on axions that may be of interest for the target students of this text too.

---

[1]We must note here that some ALP models of much higher masses (not QCD axions though) are still possible and they *can* be searched at accelerators. These searches are not considered in this review, see e.g. [1].

[2]As is customary, we will use the term axion, but often refer to other ALPs, too. When we consider it important to stress the generality of a statement we will use the term ALPs, or, conversely, we will specifically refer to "QCD axion".

## 2 Introduction to axion theory and phenomenology

Axions were originally proposed in the context of the Peccei-Quinn mechanism [7,8], to solve the *strong CP problem,* that is, the absence of charge-parity (CP) violation in the strong interactions. They were in fact identified by Weinberg [9] and Wilczek [10] as the pseudo-Nambu-Goldstone (pNG) boson of the new spontaneously broken global symmetry that Peccei and Quinn had postulated. However, the phenomenology of the axion is largely common to more generic situations involving pNG bosons with very low mass and very weak couplings coming from a spontaneously broken symmetry at very high energy scales. These axion-like particles would not be related to the PQ mechanism, and enjoy less model constraints than the "proper" (or QCD) axion. Given that solving the strong-CP problem remains a very strong theoretical motivation for these particles, let us start by briefly explaining it.

### 2.1 The strong CP problem

The Lagrangian of quantum chromodynamics (QCD), the theory that explains the strong interactions, contains the famous $\theta$-term, that violates the CP symmetry:

$$\mathcal{L}_{\cancel{CP}} = \theta \frac{\alpha_s}{8\pi} G^a_{\mu\nu} \widetilde{G}^{\mu\nu}_a, \tag{1}$$

where $\alpha_s$ is the QCD equivalent of the fine-structure constant, $G^a_{\mu\nu}$ is the gluon field-strength tensor and $\tilde{G}^{\mu\nu a}$ its dual. Viewing the QCD Lagrangian in isolation, the $\theta$ parameter can be understood as an angle determining the vacuum of the theory. However, when embedded in the full SM Lagrangian, $\theta$ receives a contribution from a transformation of the quark fields needed to remove a common phase of all quark masses (individual phase differences can be accommodated without affecting $\theta$). Because of this, it is difficult to understand why $\theta$ would be zero in the SM, in the absence of new mechanisms that somehow force it.

The $\theta$-term has no effect in perturbative calculations and that is the reason why it is often neglected. However, it has observational consequences, the most important one is the prediction of electric dipole moments (EDMs) for hadrons. In particular, the EDM expected for the neutron is:

$$d_n = (2.4 \pm 1.0)\,\theta \times 10^{-3}\,\mathrm{e\,fm}. \tag{2}$$

However, increasingly sensitive experiments have failed to detect a non-null neutron EDM, being the current most stringent upper bound [11] $|d_n| < 1.8 \times 10^{-13}\,\mathrm{e\,fm}$ (at 90% C.L.), which imposes the restriction:

$$|\theta| < 0.8 \times 10^{-10}. \tag{3}$$

The essence of the strong CP-problem is why $\theta$ is so small if composed of two phases of completely unrelated origin.

### 2.2 The Peccei-Quinn mechanism

Although some solutions to the strong CP-problem have been proposed in the literature [12, 13], including the possibility -now clearly excluded- that one of the quarks be massless, the Peccei-Quinn mechanism remains the most compelling one. The new U(1) symmetry that Peccei and Quinn postulated, now called the Peccei-Quinn (PQ) symmetry, would be spontaneously broken at a high energy scale $f_A$ [7,8]. Weinberg and Wilczek independently realised that such an spontaneously broken global symmetry implied a new pNG boson, which Wilczek

called the "axion" [14]. The low energy effective Lagrangian of the pNG of the PQ symmetry includes the term:

$$\mathcal{L}_a \ni \frac{\alpha_s}{8\pi} G^a_{\mu\nu} \widetilde{G}^{\mu\nu}_a \frac{A}{f_A},$$  (4)

which effectively replaces the $\theta$-term of the SM, absorbing $\theta$ into a redefinition of the axion field $A$[3]. The axion field now plays the role of a dynamical $\theta \to \theta(t,x) = A(t,x)/f_A$. The important point is that the potential imposed to the axion field by the QCD dynamics, in the absence of other CP-violating sources, has a minimum at the CP-conserving value $\theta = 0$. That is, the mechanism not only renders the initial $\theta$ parameter unphysical, but it dynamically settles it down to zero, effectively solving the strong-CP problem.

## 2.3 Axion mixing and mass

Some properties of the axion are determined by the PQ mechanism itself and are independent of the particular way it is implemented in the SM, i.e. they are common to all axion models. The term (4) is the defining ingredient of the PQ mechanism and implies the coupling of the axion to the gluon field. As shown by (4), the strength of this coupling is inversely proportional to the energy scale $f_A$, whose value is not fixed by theory. As will be shown below, all other axion couplings, as well as its mass, go also as $1/f_A$. Therefore, higher values of the PQ scale imply lighter and less interacting axions. The original PQWW (Peccei-Quinn-Weinberg-Wilczek) axion had $f_A$ identified with the electroweak scale. But such models were soon ruled out, as they would lead to signatures at accelerators that were not observed. Models with much larger scales $f_A$ (values well above $10^7$ GeV are needed to avoid current experimental constraints) were then proposed and dubbed "invisible axions"[4], as they were thought impossible to detect.

The term (4) also allows for the mixing of the axion with $\pi^0$ and other mesons. Through this mixing, the axion acquires a mass given by:

$$m_A = 5.70(7)\mu\text{eV}\left(\frac{10^{12}\text{GeV}}{f_A}\right).$$  (5)

Note that this mass is automatically generated by QCD, and is therefore fully determined apart from the value of $f_A$. Moreover, being a QCD effect, it vanishes for energies above the QCD scale, something that is important in cosmology, i.e. the axion is a massless particle in the early Universe. The fact that $m_A$ is univocally linked to $f_A$ through (5) means that every axion coupling is also proportional to $m_a$. Indeed, as will be shown later, axion models are typically represented as diagonal straight lines in the $(g, m_A)$ plots ($g$ being any axion coupling). Note that for generic ALPs (i.e not deriving from the PQ mechanism) this relation between $g$ and $m_a$ does not necessarily hold.

## 2.4 Axion-photon coupling

Another consequence of the mixing with mesons is a model-independent coupling to photons and hadrons. For the case of photons, the coupling has a $a$-$\gamma$-$\gamma$ form, and is the source of the axion-to-photon oscillation/conversion in the background of an electromagnetic field, a mechanism that is at the basis of important axion phenomenology. The photon interaction also allows for the decay of axions into two photons. For allowed values of $f_A$ the lifetime is

---

[3]Following [2], we will use the uppercase letter $A$ to refer to the QCD axion field, as well as to its properties ($m_A$, $g_{A\gamma}$,...), while we reserve the lowercase $a$ to refer to the more general ALP case ($m_a$,$g_{a\gamma}$,...).

[4]Now all viable models are of this kind, so the adjective "invisible" is not used. Besides, as explained here, they are at reach of current detection technologies.

much larger than the age of the Universe, so for practical purposes the axion can be considered a stable particle.

More specifically, the axion-photon interaction can be expressed with the following effective term in the Lagrangian:

$$\mathcal{L}_{A\gamma} \;\; = \;\; -\frac{g_{A\gamma}}{4} A F_{\mu\nu} \widetilde{F}^{\mu\nu} = g_{A\gamma} A \, \mathbf{E} \cdot \mathbf{B} \,, \tag{6}$$

where $g_{A\gamma}$ is the axion-photon coupling, and $F_{\mu\nu}$ the electromagnetic tensor and $\widetilde{F}^{\mu\nu}$ its dual. The equivalent term on the right expresses the interaction in terms of the electric $\mathbf{E}$ and magnetic $\mathbf{B}$ fields. It is customary to make the $\sim 1/f_A$ dependency explicit, by defining the adimensional coupling $C_{A\gamma}$:

$$g_{A\gamma} \equiv \frac{\alpha}{2\pi} \frac{C_{A\gamma}}{f_A} \,, \tag{7}$$

with

$$C_{A\gamma} = \frac{E}{N} - \frac{2}{3} \frac{4m_d + m_u}{m_u + m_d} = \frac{E}{N} - 1.92(4) \,, \tag{8}$$

being $m_u$ and $m_d$ the mass of up and down quarks respectively, and $E$ and $N$ the color and electromagnetic anomaly coefficients respectively, which depend on the particular PQ charges assigned to the particles of our theory (the axion model). Therefore the axion-photon coupling has a model independent contribution (the second term in the sum (8)) derived directly from the basic term (4), plus a model dependent one. We can thus say, barring unlikely cancellations between both terms, that the axion-photon coupling is a necessary consequence of the PQ mechanism. Due to its generality, and also to the importance of the axion-photon interaction in many of the axion detection strategies, the $(g_{a\gamma}, m_a)$ parameter space shown e.g. in Fig. 1, remains the main area to represent axion results, experimental sensitivities and observational limits. We will be referring to it often in the remainder of the report.

## 2.5 Axion models

The particular way the SM Lagrangian is completed at high energies to generate the new axion terms and the PQ mechanism, the "axion model", further determines the phenomenology of the axion, beyond the properties commented above. In particular, one has to define whether and how the particles of the SM, as well as of any extension being considered, transform under the new PQ U(1) symmetry, i.e. their PQ charges. These charges define the color and electromagnetic anomaly coefficients, $N$ and $E$, mentioned before. The original PQWW axion represented the simplest realization of the PQ mechanism in the SM, in which an extra Higgs doublet is introduced to implement the PQ symmetry, while the SM quarks are charged under the new symmetry. As mentioned above, this implementation links the scale of the symmetry to the electroweak scale, and the model was soon ruled out. Two major alternative strategies were followed to avoid this (and make the axion "invisible") that gave rise to two classes of models that are now considered as benchmarks.

- The **Kim-Shifman-Vainshtein-Zakharov (KSVZ)** model [15, 16] extends the SM field content with a new heavy quark and a singlet complex scalar. This new scalar has a potential such that the PQ symmetry is spontaneously broken with a vacuum expectation value (VEV). This VEV, $f_A$, can now be set independently much higher than the electroweak scale. In the original KSVZ model, the new fermion has no charge and

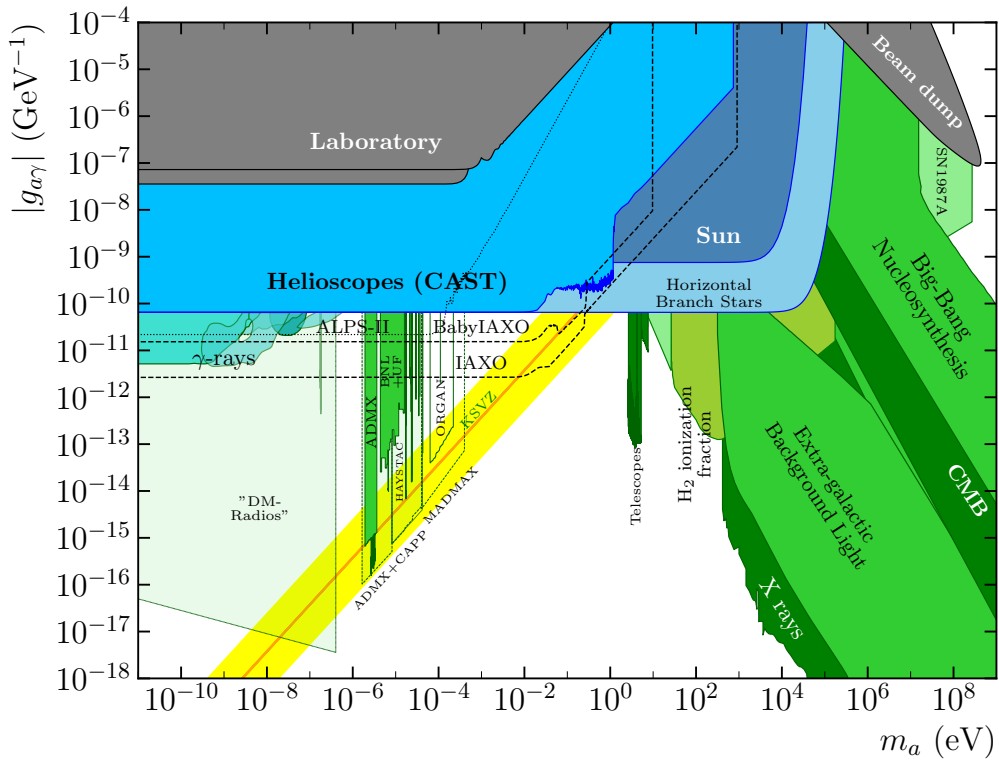

Figure 1: Overall panorama of current bounds (solid areas) and future prospects (semi-transparent areas or dashed lines) in the $g_{a\gamma}$-$m_a$ plane. See explanation in the text (mostly in sections 6 to 8) and reference [2] for details on the different lines.

the ratio $E/N = 0$. If the new heavy quark has hypercharge similar to down-type (up-type) quarks, the ratio $E/N$ equals 8/3 (2/3). KSVZ models are easily generalized to include more coloured fermions and scalars, allowing for other values of $E/N$. However, under certain requirements of stability [17], reasonable values are constrained to $E/N \in (5/3, 44/3)$. This corresponds to a span in $C_{A\gamma}$ that is represented by the yellow band shown in Fig. 1, and in other figures of this report. One of the defining features of these models are that they do not contain an axion-electron coupling at tree level (however see section 2.7). Because of this, they are sometime called "hadronic axions".

- The **Dine-Fischler-Srednicki-Zhitnitsky (DFSZ)** model [18,19], does not introduce new exotic fermions, but assigns PQ charges to the SM quarks, so that they carry the PQ anomaly. The scalar sector is however extended to contain two Higgs doublets (like in the original PQWW, to give mass to up- and down-type fermions respectively), and also a new singlet complex scalar. The latter allows to set an independent scale to the PQ symmetry. Contrary to KSVZ, these models feature axion couplings with leptons, in particular with the electron, an issue that will be commented on later. Depending on which of the Higgs are involved in the Yukawa term of the leptons, two variants of the model are possible, dubbed DFSZ-I and -II. The ratio $E/N$ is 8/3 and 2/3 for the DFSZ-I and -II respectively. Because the DFSZ models are compatible with grand unification theories (GUT) scenarios, they are sometime also called GUT axions.

Many more models have been studied in the literature, and indeed there is now an intense model-building effort in the axion phenomenology community. Many models can be considered closely related to one of the above described, but others predict axion couplings well

outside the ranges expected by KSVZ or DFSZ (we refer to the reviews [2,3] and references therein for examples). Despite this, these two classes of models remain benchmark models, and the famous yellow band shown in the figures of this report remains a major sensitivity target for experiments.

## 2.6 Axion-nucleon couplings

As mentioned above, axions feature a model-independent coupling to nucleons, derived from the mixing with mesons. However, model dependent contributions from potential axion-quark couplings may also be expected. The axion-fermion term will in general take the form:

$$\mathcal{L}_{Af} = \frac{\partial_\mu A}{2f_A} \sum_f C_{Af} \bar{f} \gamma^\mu \gamma^5 f \,, \tag{9}$$

where $f$ is the fermion field and $C_{Af}$ is the corresponding adimensional axion-fermion coupling. The low-energy couplings to neutrons $C_{An}$ and protons $C_{Ap}$ can be obtained from the quark couplings and the model-independent contributions:

$$C_{Ap} = -0.47(3) + 0.88(3)C_{Au} - 0.39(2)C_{Ad} - K_{Ah} \,, \tag{10}$$

$$C_{An} = -0.02(3) - 0.39(2)C_{Au} + 0.88(3)C_{Ad} - K_{Ah} \,, \tag{11}$$

$$K_{Ah} = 0.038(5)C_{As} + 0.012(5)C_{Ac} + 0.009(2)C_{Ab} + 0.0035(4)C_{At} \,, \tag{12}$$

where the brackets show the experimental error from quark mass estimations and NLO corrections [20], and $C_{Aq}$ with $q = d, u, s, c, b, t$ are the couplings to quarks. For the simplest KSVZ model mentioned above, all $C_{Aq} = 0$ and we are left with the model-independent contributions, while for DFSZ models:

$$C_{Au} = \frac{1}{3}\cos\beta^2 \,, \qquad C_{Ad} = \frac{1}{3}\sin\beta^2 \qquad \text{(DFSZ)} \,, \tag{13}$$

where here $u$ and $d$ refer to all up-type and down-type quarks and $\tan\beta$ is the ratio of VEVs of the two Higgs doublets in the model, and can be bounded using unitarity arguments [21] as $\tan\beta \in [0.25, 170]$.

Note that sometimes axion-fermion couplings are also expressed (in a way that for our purposes is equivalent to (9)) invoking a Yukawa-like term:

$$\mathcal{L}_{Af} = -ig_{Af} A \bar{f} \gamma_5 f \,, \tag{14}$$

where the coupling $g_{Af}$ –like the case of the axion-photon coupling $g_{A\gamma}$– is now inversely proportional to $f_A$ and can be related to $C_{Af}$ in this way:

$$g_{Af} = C_{Af} \frac{m_f}{f_A} \,. \tag{15}$$

Nucleon couplings play an important role in some stellar scenarios, and therefore are relevant to use astrophysics to contrain axion models, as will be seen below. Note that the model independent contribution to the neutron coupling is compatible with zero within errors, and that cancellations between the different parts cannot be excluded, although not simultaneously with protons *and* neutrons, at least within the simplest KSVZ or DFSZ models. Additional model ingredients can modify the above expressions and reduce hadron couplings with respect to photon couplings, the so-called astrophobic axions [17].

## 2.7 Axion-electron coupling

Axions do not couple to electrons model-independently, apart from a very small coupling that arises by radiative corrections via the photon coupling and the meson mixing, and that is usually of no practical consequence. However, specific models may feature such coupling at tree level. This is the case of DFSZ models, for which the coupling $C_{Ae}$, defined as in (9), is:

$$C_{Ae} = \frac{1}{3}\sin\beta^2\,, \qquad\qquad \text{(DFSZ}-\text{I)}\,, \qquad\qquad (16)$$

$$C_{Ae} = -\frac{1}{3}\cos\beta^2\,, \qquad\qquad \text{(DFSZ}-\text{II)}\,. \qquad\qquad (17)$$

If present, this coupling is important in some astrophysical scenarios, and therefore models featuring it are strongly constrained by astrophysics. It also allows for additional detection channels, as will be shown below.

## 2.8 Other phenomenology

The above axion couplings are the most relevant for detection but certainly not the only possible ones. For example the axion develops couplings with pions and other mesons that are relevant in cosmology. Higher dimensional terms are also possible, in particular, terms of the type $FAf f$ that leads to the existence of the neutron electric dipole moment of (2). In addition, more exotic, CP-violating scalar Yukawa couplings may be expected by e.g. new CP-violating physics beyond the SM[5] that effectively shift the minimum of the axion potential away from zero. A recent review of these couplings and how they are constrained by observations can be found in [22]. Note that if DM is composed by axions, the axion field is expected to have a local oscillating VEV, and this effectively leads to the presence of CP-violating effects, e.g. like a neutron EDM, that oscillate in time and that can be searched experimentally.

We refer to recent reviews [2,3] for a more detailed discussion on axion phenomenology.

## 2.9 Axion-like particles

The phenomenology of axions is to a large extent common to other light bosons also arising from spontaneously broken symmetries at a high energy scale $f_a$. These axion-like particles (ALPs) [23] are not in general linked to the PQ mechanism and, therefore, their mass $m_a$ and couplings $g_{a\gamma}$, $g_{ae}$, ... do not in general follow the relation with $f_a$ shown above for the axion. That is, ALPs can in general lie anywhere in the plot of Figure 1, and not just in the yellow band. For example, it is known that string theory generically predicts the existence of a large number of ALPs (in addition to the axion itself) [24–26].

Therefore it is important to consider that most axion experiments will also be sensitive to ALPs. In fact, to distinguish experimentally between a QCD axion or another type of ALP, one has to rely on the above mentioned relations between couplings and mass, and most likely more than one experimental result will be needed to confirm a positive detection as a QCD axion. Finally, a more generic category of particles called WISPs (weakly interacting sub-eV particles) also share some of the ALP phenomenology. These WISPs include, apart from ALPs, other light particles like hidden photons, minicharged particles or scalar particles invoked to explain dark energy like chameleons or galileons.

---

[5]CP-violation in the SM will also shift the minimum, but by an amount that is few orders of magnitude smaller than the current experimental bound on $\theta$.

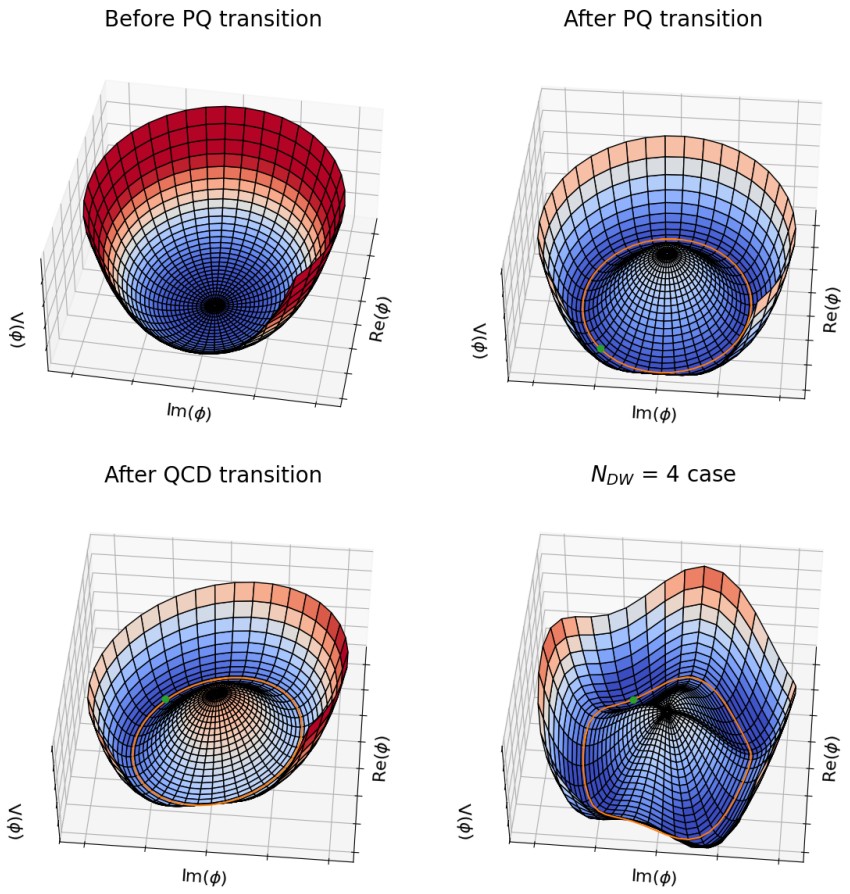

Figure 2: These figures schematically illustrate the evolution of the potential of the complex scalar field $\Phi$ whose phase is identified as the axion. When the Universe's temperature decreases below the PQ scale the minimum of the potential shifts to a non-zero value for $|\Phi|$, i.e. $V(\Phi)$ adopts the characteristic "mexican hat" shape (top-right). The energy-breaking scale corresponds to the radius of the valley from the centre. The $\Phi$ field sits at one point in the valley, i.e. it gets a VEV, and the PQ symmetry is spontaneously broken. The valley is however flat in the angular, that is, axion, dimension. Therefore, the initial value of the axion field (illustrated as the green dot) takes a random value in $[-\pi, \pi)$. At lower temperatures, QCD effects give a mass to the axion and make the potential "tilt" (bottom-left). There is now a preferred value for the axion field which turns out to be the CP-conserving value. In some models, more than one such minimum exist (bottom-right shows the $N_{DW} = 4$ case). When the temperature falls below QCD, the axion falls from wherever it was down to this minimum, and starts oscillating around it. These oscillations fill the space and behave as dark matter.

## 3 Axions in cosmology

Axions and ALPs can have an important role in many cosmological scenarios, like inflation, dark radiation, dark energy, physics of the cosmic microwave background, but, most importantly, as a potential dark matter candidate. Cosmological observations can therefore be used to constrain ALP properties, although often in a model-dependent way. We will mention some of them here, but, for the most part we will focus on the dark matter issue. We refer to specialized reviews, like [27], for additional information on axion cosmology.

Being such light particles, it may be at first sight surprising that axions could be good dark matter candidates. Indeed, thermal production of these particles in the early Universe, like in the case of neutrinos, leads to a *hot* dark matter population and therefore they do not solve the dark matter problem. However, as realized soon after their proposal (by several authors independently and simulatenously [28–30]) that the very PQ mechanism provides automatically a non-thermal production channel of a large non-relativistic population. The fact that the very axion paradigm provides a viable solution to the dark matter problem, further strengthens the axion hypothesis.

## 3.1 Axions as cold dark matter

The mechanisms through which axions may contribute to the cold dark matter density are closely connected to the PQ mechanism itself and in particular with the evolution of the axion potential with temperature. The main concept is qualitatively illustrated in Figure 2. Axions emerge first as a physically relevant degree of freedom at the PQ phase transition. Whether this transition happens before (pre-inflation scenario) or after (post-inflation scenario) inflation determines the subsequent evolution of the axion field. Later on, at the QCD scale, the axion mass "turns on" and the space is filled with a cold axion population via the vacuum realignment (VR) mechanism. In post-inflation models, an additional axion population emerges from the formation and decay of topological defects. Both mechanisms are discussed in the following. Table 1 summarizes the relevance of each production mechanism in each scenario and the main uncertainties and model dependencies.

### 3.1.1 The axion potential and the vacuum realignment (VR) mechanism

What Figure 2 shows is the potential of the Higgs-like field $\phi$ whose angle is identified as the axion ($\phi$ itself may be a different field in different axion models as explained above, but the discussion here is generic to every QCD axion model). At a very early epoch of the Universe, when its temperature crosses the PQ scale, $V(\phi)$ transitions to a characteristic "mexican hat" shape (from left to right top panels of Figure 2). That is, while first the minimum of the potential is at zero, it then moves to a non-zero value of the radial component of $\phi$, which means that $\phi$ acquires a VEV after this phase transition. At this point the potential is flat in the angular direction, so the phase of this value will take a random value around the circular "valley"[6], $\theta_i \in [-\pi, \pi]$. The PQ symmetry (which in these plots is to be seen as a rotational symmetry in the complex plane of $\phi$) is spontaneously broken.

The axion field remains massless until later times, at the moment when the temperature of the Universe reaches the QCD critical temperature. At this point the QCD effects that provide a mass to the axion become relevant. These effects can be seen as a slight "tilting" of the potential as shown on the bottom-left panel of Figure 2. Now there is a minimum $\theta_{\min}$ also in the axion potential which, as argued before, is CP-conserving and solves the strong-CP problem. However $\theta_{\min}$ will not in general coincide with the initial value $\theta_i$ in which the axion field sits just before the QCD effects are "switched on". This misalignment between $\theta_i$ and $\theta_{\min}$, which gives the name to the mechanism[7], allows for the axion to start rolling down and start performing damped oscillations around the minimum. These oscillations correspond to a coherent state of non-relativistic axions that behaves as cold dark matter (at time scales longer than the period of the oscillations) [28–30].

The density of axions produced by VR can be computed in a relatively reliable way. It requires solving the equation of motion of the axion field in the background of an expanding

---

[6]Note that the axion field and this angle are directly related, so we can also talk of a VEV of the axion field: $A_i = \theta_i / f_A$

[7]The vacuum realignment mechanism is also called sometimes axion misalignment mechanism in the literature.

Universe. The most challenging part is the calculation of the exact shape of the axion potential, especially around the critical temperature. While for small departures around the minimum $\theta_{\min}$ a harmonic approximation is accurate, for larger values the corrections for anharmonicities need to be included. Recent efforts, including lattice computations [31], have reduced QCD-related uncertainties to negligible levels when compared with other model dependencies. In summary, for a given $\theta_i$ and a given $f_A$ the density of VR axions, expressed as the ratio of the density of VR axions $\Omega_{A,\mathrm{VR}}$ over the observed total DM density $\Omega_{\mathrm{DM}}$, is [32]:

$$\frac{\Omega_{A,\mathrm{VR}}}{\Omega_{\mathrm{DM}}} \approx \theta_i^2 F\left(\frac{f_A}{9 \times 10^{11}\mathrm{GeV}}\right)^{7/6} \tag{18}$$

$$\approx \theta_i^2 F\left(\frac{6\ \mu\mathrm{eV}}{m_A}\right)^{7/6}, \tag{19}$$

where $F$ is a correction factor accounting for anharmonicities in the axion potential and other details, and itself depends on $\theta_i$ and $f_A$. As can be seen, the axion relic density is approximately inversely proportional to $m_a$, that is, the lighter the axion the higher its relic density. This is contrary to conventional thermal production like in the case of WIMPs, for which higher masses correspond to a higher dark matter density. This means that for a particular initial $\theta_i$, one can set a lower bound on the axion mass by requiring that it does not exceed the observed dark matter density. According to (19), for $\theta_i^2 F \sim 1$, the axion mass would be above $\sim 6\ \mu\mathrm{eV}$. However, much lower masses are possible if one is allowed to assume arbitrarily low values of $\theta_i$ (see later).

Obviously, the plots shown in Figure 2 describe the evolution of the axion field in a particular point in space. In general, the axion will adopt a different initial value $\theta_i$ in different causally-disconnected regions of the Universe, and it will smoothly vary between neighboring regions. If the PQ transition happens after inflation, or the PQ symmetry is restored after inflation due to reheating (the *post-inflation* scenario), the Universe remains divided in patches randomly sampling all possible values of $\theta$ with equal probability. The typical size of these patches would nowadays be $\sim 0.001(m_A/10\ \mu\mathrm{eV})^{1/2}$ pc, i.e. they are much smaller than typical cosmological probes of dark matter. Therefore the density of VR axions in the Universe can be computed using (19) but with an effective average $\theta_i = \left\langle \theta_i^2 \right\rangle^{1/2} = \pi/\sqrt{3} \simeq 1.81$, and therefore[8], the density of VR axions in the post-inflation model $\Omega_{A,\mathrm{VR}}^{\mathrm{post}}$ just depends on the axion mass:

$$\frac{\Omega_{A,\mathrm{VR}}^{\mathrm{post}}}{\Omega_{\mathrm{DM}}} \approx F\left(\frac{30\ \mu\mathrm{eV}}{m_A}\right)^{7/6}. \tag{20}$$

As can be seen, the VR mechanism in the post-inflation scenario is nicely predictive. It requires the axion mass to be about 30 $\mu$eV for it to account for the totality of DM. Unfortunately, post-inflation models need to take into account the formation of topological defects and their decay into an additional population of axions, as explained in the next section. As will be seen, this spoils the predictability of this scenario.

If the PQ transition happens during inflation, and the PQ symmetry is never restored afterwards (the *pre-inflation* scenario) then inflation selects a single $\theta_i$ patch that will be expanded to a size larger than the observable Universe, leading to a homogeneous value of the initial misalignment angle $\theta_i$. In this case, the density of VR axions is just given by (19) but the value of $\theta_i$ is unknown and can take any value $\in [-\pi, \pi)$. As mentioned above, values of $m_A$

---

[8]The presence of anharmonic corrections in the potential modifies the effective $\theta_i$ to be used in (19) to be 2.15 [20].

Table 1: Summary of the main relic axion production mechanism, their relevance in pre- or post-inflation models and main model dependencies

| Production mechanism | *pre*-inflation models | *post*-inflation models |
|---|---|---|
| Vacuum realignment (VR) | The axion DM density produced depends on the value of the initial misalignment angle $\theta_i$, which is unique for the whole observable universe, but unknown. One can fine-tune $\theta_i$ to get the desired density for a very large range of $m_a$. | $\theta_i$ takes randomly different values $[-\pi, \pi]$ in different points (patches) in the universe, so the axion density can be reliably predicted to be the one corresponding to the average $\theta_i \sim \pi/\sqrt{3}$. If this were the only production channel, the totality of DM is achieved for an axion mass of $m_a \sim 26\mu$eV. |
| Decay of topological defects (TD) | Topological defects are wiped out by inflation so they do not contribute. | Topological defects form and decay producing large amounts of axion DM. Their contribution must be computed by complex simulations, and is uncertain. Current results range from a contribution of the same order of the misalignment angle up to several times it. |
| Thermal production | Axions produced thermally (like the case of neutrinos) are relativistic, and therefore they contribute to the *hot* dark matter density. | |

much lower than the above indicated cannot be excluded if one assumes $\theta_i$ is also very small for our Universe. Given that inflation *selects* this value from a initial population of all possible values, very low $\theta_i$ for our Universe could be justified by anthropic reasons. Because of this, the window of very low $m_A$ with a finetuned low $\theta_i$ is sometimes called anthropic window.

## 3.2 Decay of topological defects (TD)

In the post-inflation scenario, the production of axions from VR is not the only cold DM production mechanism. The axion field forms topological defects, namely axion strings and walls, that subsequently decay producing additional amounts of non-relativistic axions. If the formation of defects occur during inflation, like in pre-inflation scenarios, inflation dilute them away and they do not contribute. Therefore TDs are only relevant in the post-inflation scenario.

TDs are formed via the Kibble mechanism [33] during the PQ phase transition. As mentioned above, the axion angle acquires different values in different causally disconnected patches in the Universe. When the axion mass turns on and the axion field starts rolling down to the potential minimum, it may happen that in places the axion field wraps around all the domain from 0 to $2\pi$, leaving a region in which the field is topologically trapped in the part of the domain away from the minimum, and therefore storing a huge energy density. These regions take the shape of walls and strings, and the latter may be closed or open. The walls are the boundaries between two domains in different minima. The field across the domain wall takes all values $[0, 2\pi]$ between the minima. In the strings the field takes all values $[0, 2\pi]$ along any loop enclosing the string. At the core of the string there is a singular point in which the field takes all values simultaneously, that is, the modulus of the underlying complex field vanishes. All this network of domain walls and strings that is formed during the PQ phase

transition is not stable (but for the case of $N_{\mathrm{DW}} > 1$ that is discussed below), they shrink, collide and eventually decay, radiating low momentum axions.

The computation of the density of axions produced by TD decay is difficult. Since the earliest attempts to compute it, there has been some controversy on the quantitative importance of this production mechanism, basically due to the difficulty in understanding the energy loss process of TDs and the spectrum of the axions emitted from them. Some authors argued that the contribution was of the same order as the one from the VR effect [34], while others [35] found it considerably larger. More recently, first principle field theory simulations of TDs in the expanding Universe have been attempted. These simulations are challenging because of the hugely different scales involved (e.g. thickness versus length of strings), and this requires that final results are extrapolated through several orders of magnitude in the ratio of relevant parameters. This is nowadays a very active topic of research. Recent work is shedding some light on the old controversy, but there is still a large uncertainty on the extent of its contribution to the axion cold DM density. The most recent simulation-based results point to TD axion densities about one [36–38] or even two [39,40] orders of magnitude higher than the VR one, which would raise the lower bound on the axion mass up to the $\sim$meV scale, for post-inflation scenarios (see however [41] for a skeptical view).

### 3.3 The *domain wall* problem

In some axion models, the periodical axion potential can have more than one physically distinct minimum, all of them degenerate and CP-conserving. The number of such minima is called the domain wall number, $N_{\mathrm{DW}}$. In such cases, the "tilted mexican hat" image used before is not adequate, and one should rather invoke something like what is shown on the bottom-right of Figure 2, for the case $N_{\mathrm{DW}} = 4$. That is, the circular valley of the modified mexican hat potential goes through $N_{\mathrm{DW}}$ different minima before reaching a physically equivalent value. We must note that this is not an exotic feature of some axion models, in fact, the original PQWW axion has $N_{DW} = 3$ and the DFSZ models described above have $N_{DW} = 3$ or 6.

At face value, this feature has catastrophic cosmological consequences in the post-inflation scenario. Some of the patches with different $\theta_i$ values that result from the PQ phase transition will eventually choose different minima to sit on at the QCD transition. The network of topological defects forms as described above but in this case it is stable and does not decay. With time, these TDs dominate the energy density and lead to a very different Universe not compatible with observations. Note that this problem is not present in the pre-inflation scenario, as TDs are removed by inflation. But otherwise, in the post-inflation scenario, $N_{\mathrm{DW}} > 1$ models are not cosmologically viable.

However, some interesting solutions have been proposed to solve this problem. For example, there are constructions relying on extra symmetries that feature an apparent $N_{DW} > 1$ but a physical $N_{DW}$ equal to one (we refer to [3] for an account). Another solution is to break the degeneracy of the different vacua by adding an explicit breaking of the PQ symmetry. This breaking allows for the regions in the false vacua to eventually fall to the true vacuum and allow the TDs to decay. This produces the interesting effect of making the TDs live longer than in the standard picture, resulting in higher density of TD axions. The end result is that these models can account for the totality of DM for higher axion masses.

### 3.4 Preferred $m_a$ values for axion dark matter

A very important question is whether the above considerations give any information on the axion mass, or range of masses, that are preferred for the axion to be a good DM candidate. Any such information would be precious to target experimental sensitivities. As explained above

the predicted DM density depends on $m_a$, but, unfortunately, the rest of model-dependencies prevent from obtaining clear $m_a$ targets.

In the pre-inflation models, Eq. (19) univocally links $m_a$ with the axion density $\Omega_A = \Omega_{A,\text{VR}}$ for a given initial value of the field $\theta_i$. If one requests that $\Omega_a$ equals the observed DM density, this would give a prescription on $m_a$, if it were not for the unknown value of $\theta_i$. As already mentioned, for a $\theta_i \sim 1$, we have $m_A \sim$ few $\mu$eV, but if we allow for different initial values, e.g. $\theta_i \in (0.3, 3)$, they correspond to a wider approximate range $m_A \in (10^{-6} - 10^{-4})$ eV. Even lower (or higher) finetuned values of $\theta_i$, something that could be justified by anthropic reasons [42], could lead to arbitrarily low values of $m_A$ (or as high as $10^{-3}$ eV).

In the post-inflation case, the uncertainty of an unknown $\theta_i$ is averaged away but the contribution of TDs to axion DM must be taken into account, and their calculation is complicated. As mentioned in the previous section, considerable uncertainty remains. A recent computation predicts a range for the $m_A \sim (0.6 - 1.5) \times 10^{-4}$ eV [36, 37]. Another study claims a more definite and lower prediction $m_A = 26.5 \pm 3.4$ $\mu$eV [43]. More recent work supports the high mass option, with $m_A \gtrsim 0.5$ meV (for KSVZ) and $m_A \gtrsim 3.5$ meV (for DFSZ) [40]. Arbitrarily encompassing all these results in a single range as a rough indication of the current uncertainty would give $m_A \in (0.02, 4)$ meV for the post-inflation scenario.

As mentioned above, models with $N_{\text{DW}} > 1$ are cosmologically problematic. However, those models can be made viable if the degeneracy between the $N_{\text{DW}}$ vacua is explicitly broken. In those models the topological defects live longer and produce a larger amount of axions, and therefore they can lead to the same relic density with substantially larger values of $m_A$. More specifically models with $N_{\text{DW}} = 9$ or 10 evade the contraints imposed by the argument that the breaking term should not spoil the solution to the strong CP problem, while potentially giving the right DM density for a wide $m_A \in (0.5, 100)$ meV [37].

Let us stress again that the values of $m_A$ obtained with any of the above prescriptions correspond to a $\Omega_A$ equal to the total observed DM density, and given the approximately inverse proportionality of $\Omega_A$ with $m_A$ (common for all of the axion production mechanisms discussed), lower values of $m_A$ would overproduce DM while higher masses would lead to a subdominant amount of DM.

## 3.5 ALP dark matter

All the discussion above regards the QCD axion. However, more generic ALPs can also be produced non-thermally via the realignment mechanism and contribute to the cold DM. For ALPs that couple with photons and whose $g_{a\gamma}$ is not related to $m_a$ by the model contraints of axions, a large region of the parameter space $(g_{a\gamma}, m_a)$ can provide the observed amount of dark matter [44]. As will be seen later on, this constitutes interesting targets for experiments without the sensitivity to reach QCD axion models.

## 3.6 Other cosmological phenomenology and constraints on axion/ALP properties

Cosmology itself provides opportunities to detect signals of the existence of axions or ALPs, or to produce constraints on its properties. We briefly mention some of them (we refer to the reviews mentioned in the introduction [2, 3] for additional information):

- As mentioned before, axions could also be produced thermally in the early Universe. Axions interact with pions and nucleons after the PQ transition and therefore a population would exist in thermal equilibrium with the rest of the species, and will eventually freeze out as a relic density. For axion masses in the ballpark of ~eV, such a population constitutes a hot DM component. However, the density of hot DM is constrained by cosmological observations, that can thus be used to put an upper bound on $m_A < 0.53$ eV [45].

- For the case of lighter axions, this thermally generated population behaves as dark radiation, that is, a contribution to the density of relativistic particles at the time of matter-radiation decoupling. This density is conveniently described by the effective number of neutrino species $N_{\text{eff}}$, which can be measured via cosmological observations of the CMB or the large-scale structure of the Universe. Our current best determination is $N_{\text{eff}} = 2.99 \pm 0.17$ [46], compatible with the SM expectations $N_{\text{eff}}^{\text{SM}} = 3.045$ [47]. If the axion thermalization happens above the electroweak scale we expect an additional contribution of at least $\Delta N_{\text{eff}} \sim 0.027$, although higher values are possible for other model-dependent couplings. This value is small, but future cosmological probes will be able to be sensitive to it [48]. It is particularly interesting that the current tension between the early and the late Universe determination of the Hubble constant [49] can be alleviated by a hot axion component of the kind here discussed [50].

- In the pre-inflation scenario, the axion exists during inflation and its quantum fluctuations are expanded to cosmological sizes, contributing to the temperature inhomogeneities of the cosmic microwave background (CMB) with so-called isocurvature fluctuations. The absence of this signal in CMB observations can be translated into constraints on the axion DM density that is however dependent on $\theta_i$ and $H_I$, the expansion rate of inflation, currently unknown. Let us note however that a measurement of $H_I$ is possible in next generation CMB polarization experiments if they find B-modes from primordial gravitational waves during inflation. Such a discovery would likely rule out completely the pre-inflation scenario, as was thought to happen after the BICEP2 claim [51] a few years ago, later retracted.

- Even if the decay of axions (or ALPs) into photons is longer than the age of the Universe, it may still have observable consequences. In DM rich regions, a monochromatic emission of gammas at an energy equal to half the axion mass would be expected. Such a line has been searched for in visible wavelengths, giving rise to an exclusion labelled "telescopes" in Figure 1 (see, e.g. [52] for the most recent one). Searches in the microwave regime have been carried out, but with less sensitivity. However, the option of invoking induced decay (inverse-Primakoff conversion) by intervening strong galactic $B$-fields like the ones around neutron stars has allowed to probe relevant ALP DM regions [53–56]. Finally, ALP DM decay has been proposed to explain the 3.55 keV line that is observed in some galaxy clusters [57, 58].

- Shorter decay times would have other cosmological consequences, like distortion of the CMB spectrum, affect the result of the primordial nucleosynthesis, produce monochromatic X-ray and gamma-ray lines in the extragalactic background light, or alter the $H_2$ ionization fraction. We refer to [59] for more details of these arguments that lead to some of the constraints shown in green in the bottom-right corner of Figure 1.

- The physics of the inflaton field – the hypothetical field that drove inflation in the early Universe – has some similarities with the axion potential and many attempts have been done to embed inflation into an axion/ALP framework. The inflaton has been identified with the axion itself, the radial mode of the PQ complex field, or a combination of the latter with another Higgs-like field (we refer to [3] for an account of those models). In general, the predictions of these models are difficult to test experimentally. A possible exception is the so-called "ALP-miracle model" of [60, 61] where an ALP can both drive inflation and provide the DM of the Universe through the realignment mechanism with a mass in the range $\sim 0.01 - 1$ eV, and values of $g_{a\gamma}$ at the reach of current experiments.

- There is an important consequence of the VR mechanism in the post-inflation scenario. Even if the average VR axion density is quantified by Eq. (20), the actual density will

be quite inhomogeneous due to the different initial $\theta_i$ values adopted by the axion field after the PQ transition in different patches of the Universe. Regions with an initial over-density will become gravitationally bound and collapse forming relatively dense axion miniclusters [62,63]. Their typical size and mass have been computed for the QCD axion to be of the order $R_{mc} \sim 2.5 \times 10^8$ km and $M_{mc} \sim 10^{-11} M_\odot$, respectively (where $M_\odot$ represents one solar mass). This could have important consequences for direct detection experiments. An encounter of the Earth with an axion minicluster could enhance the local DM density by a factor $10^6$, but only for a short time. Which fraction of the axion DM is in the form of miniclusters is being studied via simulations. In addition, the presence and amount of miniclusters, could also be assessed with future micro-(or pico-)lensing observations.

- Since the detection of the first gravitational waves (GW), the possible options to hint at the presence of cosmological axions in terms of GW signals has become an important topic of study. It has been recently pointed out that in post-inflation models and sufficiently low axion mass, the formed topological defects produce a contribution to the stochastic gravitational wave background that could be observable [64]. In addition, ALPs models with long-lived topological defects ($N_{DW} > 1$ as discussed above), could produce observable GW for a large range of axion mass, if the defects decay late enough [65].

- Theoretical efforts to explain the identity of dark energy have introduced scalar fields that, although not strictly being ALPs, they share in some cases similar phenomenology. Some examples are quintessence fields [66–69], chameleons [70–72], galileons [73], symmetrons [74]. Particularly relevant is the case of chameleons, that have been searched for as a byproduct of axion experiments [75–78].

# 4 Axions in astrophysics

Being such light particles, and by virtue of their interaction with photons, electrons and nucleons, axions and ALPs may have important effects in the evolution of stars. They can be produced in the stellar interior, and like neutrinos, due to the smallness of their interaction, they can easily escape the star. So they may constitute an efficient mechanism of energy drain and they can alter the lifetime and other features of the star. This fact has been used to constrain axion properties, and indeed the most stringent bounds on most ALP couplings come from astrophysical considerations. Although some calculations have been updated since its publication, the classic book by G. Raffelt [79] is still a great reference to review the role of these particles in the stellar environments.

In the next subsections we review the main results in this respect, with a particular emphasis on observations that, instead of constraining the properties of the ALPs, they seem to hint at them. Later on we review another astrophysical scenario in which ALPs can play a relevant role: the propagation of gamma-rays in galactic or intergalactic magnetic fields.

## 4.1 ALPs and axions in stellar evolution

The different stages of the life of a star are associated to the type of nuclear fuel being burnt in its interior, i.e. young stars obtain their energy by fusion of Hydrogen into Helium, later Helium into Carbon and so on with heavier elements. Each stage also has associated a region in the famous Hertzsprung-Russel (HR) or colour-magnitude diagram that shows the luminosity and surface temperature of the star (e.g Hydrogen burning stars in the "main sequence", Helium

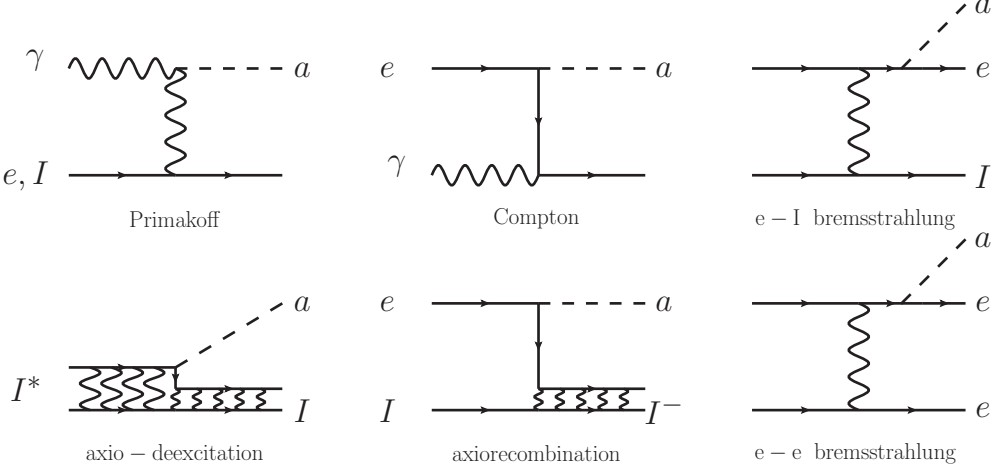

Figure 3: Feynman diagrams of some of the processes producing axions in the stellar interiors. The Primakoff conversion of photons in the electromagnetic fields of the stellar plasma depends on the axion-photon coupling $g_{a\gamma}$, and is present is practically every axion model. In non-hadronic models, in which axions couple with electrons at tree level, additional mechanisms are possible, like: atomic axio-recombination and axion-deexcitation, axio-Bremsstrahlung in electron-ion or electron-electron collisions and Compton scattering with emission of an axion. Collectively, the flux of solar axions from all these latter $g_{ae}$-mediated channels is sometimes called ABC axions, from the initials of the mentioned processes. In the diagrams, the letters $\gamma$, $a$, $e$ and $I$ represent a photon, axion, electron and ion respectively. Figure from [82].

burning stars in the "horizontal branch",...). Throughout its life, each star evolves in the HR diagram in a way that depends on its initial mass, but is otherwise dictated by the nuclear physics involved. The measurement of the distribution of stars in the HR diagram allows for the reconstruction of the evolutionary time of each of the stages. Numerical simulations reproduce the HR distribution of stellar populations remarkably well and can be used to constrain the presence of new physics. For example if a new mechanism of energy drain is present due, e.g., to the production of axions at a particular stage of the star's lifetime, it will lead to a shortening of the time of this particular stage. This will show as a reduction in the amount of stars observed in the corresponding region of the HR diagram, with respect to the predictions of the standard stellar models. Depending on the ALP production mechanism that is relevant in the particular nuclear environment of the star, a different ALP coupling may be probed by different stars at different evolutionary stages.

### 4.1.1 ALP-photon coupling

Axions can be produced in the stellar interiors by the Primakoff conversion of thermal photons in the electrostatic field of electrons and nuclei (see top-left diagram in Figure 3). This process is more important in hot (as the number of thermal photons increases), but not very dense (to avoid high plasma frequency), stellar interiors. This is the case of the Sun, for which the Primakoff axions are a prime target for direct detection, as will be seen in section 8. The presence of such an exotic cooling process in the Sun would have an impact in its lifetime, but most sensitively in helioseismological observations and in the measured solar neutrino flux. These two observations have been used to constrain the ALP-photon coupling as $g_{a\gamma} \leq 4.1 \times 10^{-10}$ GeV$^{-1}$ (at $3\sigma$) [80] and $g_{a\gamma} \leq 7 \times 10^{-10}$ GeV$^{-1}$ [81], respectively.

But the strongest bound on $g_{a\gamma}$ is achieved using horizontal-branch (HB) stars. These

are Helium burning stars, with a low core density and high temperatures. Axions could be efficiently produced via Primakoff conversion and speed up the evolution of the star in this stage. Observationally, the relevant parameter is the ratio of stars in the HB stage over the ones in the Red Giant Branch (RGB), the stage just preceding the HB, known as the *R*-parameter. The presence of a non-zero $g_{a\gamma}$ will reduce *R*, that is, will deplete the stars at HB with respect to the RGB ones. From measurement of *R*, the strongest bound on $g_{a\gamma}$ can be obtained [83], known as the HB bound:

$$g_{a\gamma} < 0.66 \times 10^{-10} \text{GeV}^{-1} (95\% \text{CL}). \tag{21}$$

In the same work [83], the value determined for *R* is a bit smaller than expected, leading to the preference of a small, non-vanishing $g_{a\gamma}$, a result known as the "HB hint":

$$g_{a\gamma} = (0.29 \pm 0.18) \times 10^{-10} \text{GeV}^{-1} (68\% \text{CL}). \tag{22}$$

Note that the presence of axion-electron processes would also produce a similar result, and therefore there could be a degeneracy of the effects of both $g_{a\gamma}$ and $g_{ae}$. The results above (21) and (22) assume the axion-electron coupling can be neglected.

Further evidence for a non-zero $g_{a\gamma}$ have been suggested in the literature. Heavy stars in the He-burning stage have a particular evolution towards the blue (hotter) region of the HR diagram and back, a feature known as the *blue loop*. The time spent in this transient would be particularly sensitive to $g_{a\gamma}$ values at the level or somewhat smaller than the bound (21), and indeed such a case could explain the observed deficit of blue versus red stars [84]. A different observation is that surveys show that the SN type II progenitors are red supergiants with a certain maximal luminosity. This restriction is not understood with standard stellar models and an exotic cooling mechanism like the ones discussed here could help reconcile observations with simulations [85].

### 4.1.2 ALP-electron coupling

If the axion couples with electrons, a number of additional axion production mechanisms are at play in dense stellar interiors, namely: atomic axio-recombination and axion-deexcitation, axio-Bremsstrahlung in electron-ion or electron-electron collisions and Compton scattering with emission of an axion, whose Feynman diagrams are shown in Figure 3. Collectively, these $g_{ae}$-mediated mechanisms are sometimes called ABC processes, from the initials of the mentioned processes. In the Sun, ABC axions offer an interesting detection possibility in axion helioscopes, as will be discussed in section 8. Regarding the possibility to constrain $g_{ae}$, the most interesting options are the dense cores of white dwarfs (WD) and RGB stars, for which the bremsstrahlung emission dominates.

WDs are relatively light stars in a late stage of their lifetime, when they have exhausted their nuclear energy sources. Then the evolution of the star follows a simple well-understood gravothermal process, governed by the cooling offered by photon and neutrino emissions. The presence of an exotic cooling mechanism could be made evident in two different ways. The first one is in the shape of the WD luminosity function (WDLF), that is the distribution of WDs versus luminosity. The most complete measurements of the WDLF, using populations of the order of $10^4$ stars, find a slight disagreement with calculations, and favor the hypothesis of an additional cooling mechanism at $\sim 2\sigma$. The result has been reproduced in several studies [86, 87]. When interpreted as a hint for a non-zero value of $g_{ae}$ the following range is obtained:

$$g_{ae} = (1.5^{+0.6}_{-0.9}) \times 10^{-13} \quad (95\% \text{ CL}). \tag{23}$$

An independent method to confirm the existence of such an exotic cooling mechanism is offered by the direct observation of the period change of single WD variable stars, i.e. WDs whose luminosity oscillates due to gravity pulsations within themselves. This allows to measure the cooling rate directly for that particular star. However, due to the slow rate of period change, very long observations (decades apart) are needed, and they are available for just a few stars [88]. Interestingly, for all those cases the rate of change measured is larger than expected, hinting at an additional cooling channel. When interpreted as $g_{ae}$-mediated axion production, they point to values in the ballpark of a few $10^{-13}$.

Another good observable to constrain $g_{ae}$ is the luminosity of the RGB tip, the point of maximum luminosity, when RBG stars reach the condition to ignite Helium (known as the He-flash). This observable has been originally studied for two globular clusters, M5 [89] and M3 [90], but more recently extended to many more clusters and with better data quality in distance determination [91,92] providing an upper limit:

$$g_{ae} < 1.3 \times 10^{-13} \quad (95\% \text{ CL}). \tag{24}$$

The statistical combination of the results from the WDLF, the WD pulsation and the RGB stars favors the axion solution with slightly more than $\sim 3\sigma$, providing a good fit and a best fit range [93]:

$$g_{ae} = (1.6^{+0.29}_{-0.34}) \times 10^{-13} \quad (1\sigma). \tag{25}$$

These hints can also be combined with the *R*-parameter results discussed before, taking into account that the latter can also be explained by a non-zero $g_{ae}$, leading to a hinted region in the combined $(g_{a\gamma}, g_{ae})$ plane [93]. It is remarkable that it is in part compatible with QCD axion models with masses in the few meV ballpark[9].

### 4.1.3 ALP-nucleon coupling

If axions or ALPs couple to nucleons, it allows for nuclear transition in the stellar core to emit axions. In the Sun, this emission has been searched for in experiments (see section 8). More relevant from the standpoint of stellar evolution are thermal processes like nucleon bremsstrahlung. This process is efficient at temperatures high enough so that the momentum exchange between the nucleons is larger than the pion mass. This happens only at the cores of supernovae (SN) and neutron stars (NS).

Indeed the strongest constraint on the axion-nucleon interaction comes from the famous observation of the neutrino signal from the supernova explosion SN1987A. The signal duration depends on the efficiency of the cooling and the observed spread in the few neutrinos detected is compatible with the standard picture that neutrinos dominate as the carrier of the energy released during the explosion. This can be used to put a constraint on any additional exotic energy loss mechanism like the one offered by an axion-nucleon interaction. The most recent analysis [94] leads to the combined bound on the axion-proton $g_{ap}$ and axion-neutron $g_{an}$ interactions:

$$g_{an}^2 + 0.61 g_{ap}^2 + 0.53 g_{an} g_{ap} \lesssim 8.26 \times 10^{-19}. \tag{26}$$

---

[9]The hint of Eq. (25) includes an old version of the RGB tip analysis, and is now in the process of being updated with the result quoted in Eq. (24), although it is not expected to change by much. M. Giannotti, private communication.

We must stress that considerable uncertainty remains in the derivation of this bound, that stems from the supernova modelling itself, the sparse neutrino data on which it is based, and from the difficulty of describing the axion production in processes in a high density nuclear medium. Regarding the latter, there is a long track of studies reconsidering the modelling of such nuclear processes. We refer to [3] and references therein for more information. For QCD axions, sometimes this bound is expressed as a very stringent upper bound on the axion mass $m_A \lesssim 20$ meV. To the above caution one has to add the model dependencies linking $g_{An}$ or $g_{Ap}$ to $m_A$. As mentioned in [32], this limit must be considered as indicative rather than a sharp bound.

The observed cooling rate in some NS have also been used to constrain axion-nucleon couplings [95–97]. In this case also a possible hint of extra cooling was suggested [98], however it was later explained by standard processes [99]. In general the constraints from NS are similar to the ones above from SN, and the same words of caution apply.

## 4.2 ALPs and the propagation of photons over large distances

ALP-photon conversion (and viceversa) can also take place in astrophysical magnetic fields, by virtue of the same Primakoff conversion that is invoked in many of the detection techniques discussed later on. For a monochromatic beam of photons traversing a homogeneous magnetic field, the effect can be seen as a mixing between the photon and the ALP, giving rise to photon-ALP oscillations similar to the well know neutrino oscillations. In general, the final result of these oscillations can be rather complex, and depends on the energy spectrum of the photons and the distribution of the magnetic field. If the field extends over large distances, the conversion probability gets enhanced by coherence effects, if the axion mass is low enough. Therefore, even in the relatively low intergalactic magnetic fields, relevant effects are possible. For high-energy gammas propagating large distances, two such effects can be observable: 1) oscillatory features in the photon spectrum detected at Earth, due to some photons converting to ALPs and viceversa, where the particular shape of these oscillations will depend on the morphology of the intervening magnetic field, and 2) a boost in the photon flux due to the reconversion of photons from ALPs that effectively reduces the opacity of the medium to the photons (that is, the astrophysical version of the "light shining through a wall" experiment discussed in section 6). Both effects have been searched for in observations of distant sources of high-energy photons, and constraints (and in some cases hinted regions) on the ALP properties have been derived.

Photons traversing large distances in the Universe may interact with the low energy photons of the extragalactic background light (EBL) producing electron-positron pairs. If the photon mixes with the ALP, which does not interact with the EBL, the effective optical depth is larger than the one evaluated by conventional physics. The EBL is the background radiation field which encompasses the stellar emission integrated over the age of the Universe, and the emission absorbed and re-emitted by dust. However, it is difficult to measure directly and only indirectly-constrained models exist with considerable uncertainties.

A number of works have found evidence that current EBL models over-predict the attenuation of gamma rays using published data points of Active Galactic Nuclei (AGN) spectra obtained with imaging air Cherenkov telescopes (IACTs), which measure gamma rays above energies of ∼50 GeV. Such an over-prediction would manifest itself through a hardening of the AGN spectra, i.e. increasing the exponent of the power-law typically describing such spectra. Such increase seems to be correlated with optical depth, something that favors the interpretation of photon-ALP mixing. Different variations of this scenario have been studied in the literature, depending on which magnetic field is relevant to the mixing, e.g. the extragalactic field, or the fields at the origin (in the AGN itself) or at the end (the Milky Way field). Some authors have quantified such interpretation into particular "hinted" regions in the ALP param-

eter space (see e.g. "T-hint" labelled regions in Figure 7) in some cases up to 4-$\sigma$ claimed. For the ALP to account for these effects, very low masses at around $m_a \sim 10^{-8-7}$ eV and couplings $g_{a\gamma} \sim 10^{-11-10}$ GeV$^{-1}$ are needed, values that are not compatible with QCD axions. However, the effect has not been reproduced in other studies, when taking into account experimental uncertainties. In addition, alternative interpretations based on standard physics have been proposed in the literature. Some of the "negative" works have produced excluded regions that partially cancel the hinted ones (see "HESS", "Mrk421" or "Fermi" -labelled regions in the same figure). In any case, whether there still is a hint for ALPs in these observations or not remains a controversial issue. We refer to section 7 of [100] for a balanced discussion of this issue and a list of relevant references. Fortunately, the relevant region will soon be probed experimentally, as discussed later on in sections 6 and 8 .

## 4.3 Other astrophysical phenomenology

Other astrophysical scenarios different from the ones above described have been studied in regards to possible signals or constraints to ALPs and axions. Some of them are briefly mentioned here:

- For very light $m_a$, with Compton wavelengths comparable with the radius of a black-hole, the latter can efficiently lose angular momentum into ALPs [101]. This is the phenomenon called *blackhole superradiance*. Therefore the existence of blackholes with large angular momentum can be used to strongly disfavour ALPs. This argument excludes ALPs in the band $6 \times 10^{-13}$eV $< m_a < 2 \times 10^{-11}$eV [102], as well as other ranges at lower values [103].

- ALPs could also be produced during the core collapse of supernovae in the electrostatic fields of ions and escape the explosion. If they convert back into gamma rays in the Galactic magnetic field, a gamma ray burst lasting tens of seconds could be observed in temporal coincidence with the SN neutrino burst. The non observation of such a gamma-ray burst from SN1987A leads to a constraint to $g_{a\gamma}$ at low masses (see SN1987A labelled region in Figure 7) [104,105] (although see [106] for a critical comment on this bound). Interestingly, if a Galactic SN occurred in the field of view of the Fermi LAT, a wide range of photon-ALP couplings could be probed for masses below 100 neV. The diffuse SN flux, i.e. the cumulative emission of ALPs from all past SN explosions, have also been used to constrain ALP properties [107].

- X-ray astronomy observations have been used to search for X-ray emissions coming from the conversion of stellar ALPs in the Galactic field, both from single stars [108], or from dense stellar clusters [109]. Spectral distortions similar to the ones described above for gamma sources have been studied and searched for in X-ray point sources in galaxy clusters. The absence of such distortions was used to constrain $g_{a\gamma} < 1.5 \times 10^{-12}$ GeV$^{-1}$ at very low masses $m_a < 10^{-12}$ eV [110,111].

## 5 Detecting low energy axions

Because of what has been described in the previous sections, the search for axions is nowadays a very motivated experimental goal. Axions are expected to be very light particles and therefore signals of their existence are not expected at accelerators. More generic ALPs may evade the constraints on the axion mass and relatively massive ALP models are still viable. These models can still be searched for at accelerators [1], but they will not be treated here. In the rest of this text, we will refer to detection strategies for *low energy* axions and ALPs, where

low energy means $m_a \lesssim 1$ eV. The search for such low energy axions represents a particular experimental field that requires very specific combinations of know-hows, some of them not present in typical HEP groups, and therefore requiring cross-disciplinary technolgy transfer. They include, among others, high-field magnets, super-conduction, RF techniques, X-ray optics & astronomy, low background detection, low radioactivity techniques, quantum sensors, atomic physics, etc. Their effective interplay with axion particle physicists is an important challenge in itself, that will be conveyed in the following sections.

We describe in the following the strategies to *directly* detect axions in laboratory experiments, being these axions produced in the laboratory itself or coming from other natural sources (in contrast to indirect detection, or detection of signatures of axions in cosmology or astrophysics like the ones described in the previous sections). The most relevant sources of axions are the Sun and the dark matter halo. It is customary then to categorize the different experimental approaches according to the source of axions used:

- Experiments looking for axions or axion-induced effects produced and/or detected entirely in the laboratory.

- Experiments attempting the detection of the very axions that constitute our local dark matter galactic halo, often called "axion haloscopes" [10]

- Experiments searching for axions emitted by the Sun and detected at terrestrial detectors, or "axion helioscopes".

Purely laboratory-based experiments constitute the most robust search strategy, as they do not rely on astrophysical or cosmological assumptions. However, their sensitivity is hindered by the low probability of photon-axion-photon conversion in the lab. Haloscopes and helioscopes take advantage from the enormous flux of axions expected from extraterrestrial sources. Because of this, they are the only techniques having reached sensitivity down to QCD axion couplings. Haloscopes rely on the assumption that the 100% of the dark matter is in the form of axions, and in the case of a subdominant axion component their sensitivity should be rescaled accordingly. Helioscopes rely on the Sun emitting axions, but in its most conservative channel (Primakoff conversion of solar plasma photons into axions) this is a relatively robust prediction of most models, relying only on the presence of the $g_{a\gamma}$ coupling.

As will be shown in the following, most (but not all) of the axion detection strategies rely on the axion-photon coupling $g_{a\gamma}$. This is due to the fact that this coupling is generically present in most axion models, as well as that coherence effects with the electromagnetic field are easy to exploit to increase experimental sensitivity. The three forthcoming sections briefly review the status of the three experimental "frontiers" above listed. Fig. 1 shows the overall panorama of experimental and observational bounds on the $g_{a\gamma}$-$m_a$ plane. Some of the latter have been commented in the previous sections, for a more detailed description, we refer to [2].

# 6 Axions in the laboratory

The most well-known technique to search for ALPs purely in the laboratory is the photon regeneration in magnetic fields, colloquially known as *light-shining-through-walls* (LSW). A powerful source of photons (e.g. a laser) is used to create axions in a magnetic field. Those

---

[10]The name *axion haloscopes* (as in the case of *axion helioscopes*) was coined by P. Sikivie in his seminal paper [112] in which the –now widely spread– magnetized RF-cavity approach was first proposed. Nowadays many variations of this method, or altogether new approaches, are being followed. The name *axion haloscope* is sometimes used extensively for any technique looking for DM axions, and other times restricted to the conventional Sikivie haloscopes.

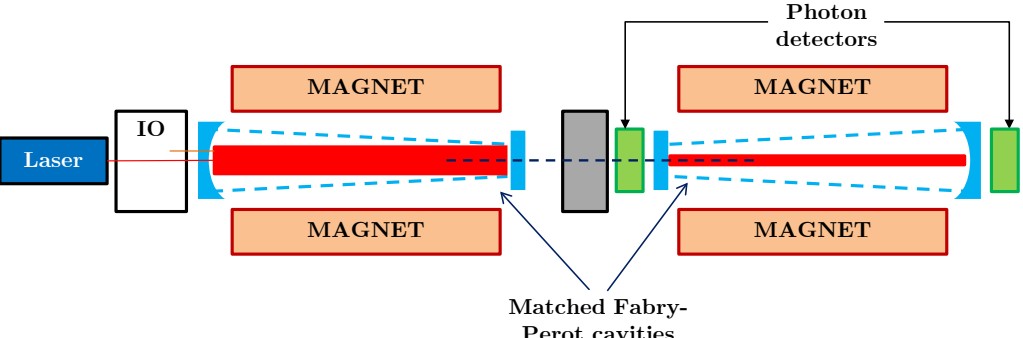

Figure 4: The principle of photon regeneration. The laser on the left injects a large number of photons to the production region. Some of them convert into axions that traverse the opaque wall in the middle into the regeneration region on the right. Photon produced by the back conversion of these axions in this second region are detected by appropiate low-noise sensors. The resonantly enhanced version includes Fabry-Perot cavities to increase the probability of conversion. The cavities in both the production and regeneration regions must be actively locked in order to gain in sensitivity. Figure from [2].

axions are then reconverted into photons after an optical barrier. Other techniques in this category are the search for alterations in the polarization of laser beams traversing magnetic fields, or the presence of new macroscopic forces that could be mediated by these particles.

## 6.1 LSW experiments

Figure 4 shows the conceptual arrangement of LSW experiments. The left half is the *production* region, where photons from the source are converted into axions. The right half is the *reconversion* region, where axions are converted into photons, that are subsequently detected. In the resonantly enhanced version of a LSW experiment long optical cavities (i.e. Fabry-Perot resonators) are placed in the production and maybe also in reconversion regions, in order to boost the conversion probability. The two resonators must be mode-matched and phase-locked, which is technologically challenging.

As already mentioned, the axion mixes with photons when propagating in a magnetic field, and the result can be interpreted as an axion-photon oscillation similar to the well-known neutrino oscillations. In the limit of small mixing, and relativistic velocities, the probability of an axion to convert into a photon (or viceversa) after traversing a length $L$ in a homogeneous magnetic field $B$ (perpendicular to the propagation direction) can be expressed as:

$$\mathcal{P}(\gamma \to a)(L) = \mathcal{P}(a \to \gamma)(L) = \left(\frac{g_{a\gamma}B}{q}\right)^2 \sin^2\left(\frac{qL}{2}\right), \tag{27}$$

where $q \sim m_a^2/2\omega$ is the momentum difference between the photon and axion waves, and $\omega$ being the energy of the axion/photon. If $qL \ll 1$, i.e. when the length $L$ is much smaller than the characteristic oscillation length, the probability becomes proportional to $L^2$:

$$\mathcal{P}(\gamma \to a)(L) \sim \left(\frac{g_{a\gamma}BL}{2}\right)^2. \tag{28}$$

In a LSW experiment the double conversion $\gamma \to a \to \gamma$ must take place to produce a signal in the detector, and therefore the double conversion probability is what is relevant for

the figure of merit of the experiment:

$$\mathcal{P}(\gamma \to a \to \gamma) = \mathcal{P}(\gamma \to a)\mathcal{P}(a \to \gamma) \sim \left(\frac{g_{a\gamma}BL}{2}\right)^4 \beta_P \beta_R, \qquad (29)$$

where now we have added the factors $\beta_P$ and $\beta_R$, that are the power built-up factors of the production and regeneration cavities respectively, only relevant in the resonantly enhanced version of the experiment. They account for the quality of the resonators, and can be understood as the times the laser bounces back and forth between the mirrors increasing the chance of conversion[11]. Expression (29) assumes equal length $L$ for both production and regeneration regions. As in previous expressions, coherent conversion is assumed, which means that the sensitivity of these experiments is independent of the axion mass until $qL \ll 1$ is not true anymore. Above this point the sensitivity drops, determining the characteristic shape of LSW exclusion lines in the $(g_{a\gamma}, m_a)$ plane, which are flat in $m_a$ until a value above which the line quickly goes up.

A number of LSW experiments have been carried out in the past [114], all of them producing limits to $g_{a\gamma}$ in the ballpark of $10^{-6} – 10^{-7}$ GeV$^{-1}$. Currently two active collaborations are working on LSW experiments and have produced the most competitive bounds below $10^{-7}$ GeV$^{-1}$: The ALPS [115] experiment at DESY and the OSQAR [116] experiment at CERN, both make use of powerful accelerator dipole magnets, from HERA and LHC accelerators respectively. ALPS enjoys power build-up in the production region, while OSQAR has slightly higher magnet and laser parameters. In both cases the bound is valid for $m_a \sim 10^{-4}$ eV above which the coherence is lost and the sensitivity drops.

These results can be improved by implementing resonant regeneration schemes. If adequately matched Fabry-Perot resonators are used in both the generation and conversion parts, improvement factors $\beta_P \beta_R$ of several orders of magnitude can be obtained. However, it poses challenging requirements on the optical system. The ALPS II experiment [117], currently finishing construction at DESY, will be the first laser LSW using resonant regeneration in a string of 2×12 HERA magnets (i.e. a length of 2×120 m) for the production and the conversion regions. The expected sensitivity of ALPS II goes down to $g_{a\gamma} < 2 \times 10^{-11}$ GeV$^{-1}$ for low $m_a \lesssim 10^{-4}$ eV, and will be the first laboratory experiment to surpass current astrophysical and helioscope bounds on $g_{a\gamma}$ for low $m_a$, partially testing ALP models hinted by the excessive transparency of the Universe to ultra-high-energy (UHE) photons (see Fig. 1). A more ambitious extrapolation of this experimental technique is conceivable, for example, as a byproduct of a possible future production of a large number of dipoles like the one needed for the Future Circular Collider (FCC). This is the idea behind JURA, a long-term possibility discussed in the Physics Beyond Colliders study group [118]. JURA contemplates a magnetic length of almost 1 km, and would suppose a further step in sensitivity of more than one order of magnitude in $g_{a\gamma}$ with respect to ALPS II.

LSW experiments with photons at frequencies other than optical have also been performed. The most relevant result comes from the CROWS experiment at CERN [119], a LSW experiment using microwaves [120]. Despite the small scale of the experiment, its sensitivity approached that of ALPS or OSQAR, thanks to the resonant regeneration, more easily implemented in microwave cavities. A large-scale microwave LSW experiment has been discussed in the literature [121]. LSW experiments have also been performed with intense X-ray beams available at synchrotron radiation sources [122,123]. However, due to the relative low photon number available and the difficulty in implementing high power built-ups at those energies,

---

[11]Note that the power build up of the regeneration cavity also contributes even if obviously there is no axion bouncing back and forth the mirrors. This can be understood noting that the axion drives the reconversion cavity at the resonant frequency. This is known as "resonant regeneration" and is also similar to the Purcell effect, as noted in [113].

X-ray LSW experiments do not reach the sensitivity of optical or microwave LSW.

## 6.2 Polarization experiments

Laser beams traversing magnetic fields offer another opportunity to search for axions. The photon-axion oscillation in the presence of the external $B$-field has the effect of depleting the polarization component of the laser that is parallel to the $B$-field (dichroism), as well as phase-delaying it (birefringence), while leaving the perpendicular component untouched. The standard Euler-Heisenberg effect in QED (also dubbed *vacuum magnetic birefringence*) would be a (still unobserved) background to these searches. The most important experimental bound from this technique comes from the PVLAS experiment in Ferrara [124], reaching a sensitivity only a factor of $\sim 8$ away from the QED effect [125]. The BMV collaboration in Toulouse [126], as well as OSQAR at CERN have reported plans to search for the QED vacuum birefringence. Recently, efforts towards an enhanced experiment of this type, dubbed VMB@CERN, are being discussed in the context of the Physics Beyond Colliders initiative at CERN.

## 6.3 New long-range macroscopic forces

Although very different from the above examples, experiments looking for new macroscopic forces (e.g. torsion balance experiments, among many others) could in principle be sensitive to axion effects in a purely laboratory setup. Axion-induced forces via e.g. a combination of axion-fermion couplings, could compete with gravity at $\sim 1/m_a$ scales. However, the interpretation of current bounds in terms of limits to axion couplings are typically not competitive with astrophysical bounds or electric dipole moment (EDM) limits on CP-violating terms (see [2] for a recent discussion). The recently proposed ARIADNE experiment intends to measure the axion field sourced by a macroscopic body using nuclear magnetic resonance (NMR) techniques [127] instead of measuring the force exerted on the other body. Very relevantly, ARIADNE could be sensitive to CP-violating couplings well below current EDM limits, in the approximate mass range 0.01 to 1 meV. Therefore, it could be sensitive to QCD axion models with particular assumptions; most importantly, they should include beyond-SM physics leading to a CP-violating term much larger than the expected SM contribution. Because of this assumption, ARIADNE would not allow for a firm model-independent exclusion of the axion in this mass interval.

# 7 Dark matter experiments

If our galactic dark matter halo is totally made of axions, the number density of these particles around us would be huge. The density of local dark matter is measured to be about $\rho \sim 0.2 - 0.56\,\text{GeV/cm}^3$ [128], which means that we would expect an axion number density of the order of:

$$n_a \sim \rho_a/m_a \sim 4 \times 10^{13} \left( \frac{10\,\mu\text{eV}}{m_a} \right) \text{axions/cm}^3 . \tag{30}$$

These DM axions would be non-relativistic particles, with a typical velocity given by the virial velocity inside our galaxy, $\sim 300$ km/s (that is, the axions get their velocity mainly by falling in the galactic potential well). The precise velocity distribution around this value is however dependent on assumptions on how the Milky Way dark matter halo formed. The typical approach in experiments is to follow the Standard Halo Model (SHM), which comes from the simplistic assumption that the halo is a thermalized pressure-less self-gravitating sphere of particles. The velocity distribution of the SHM at the Earth is given by a Maxwellian

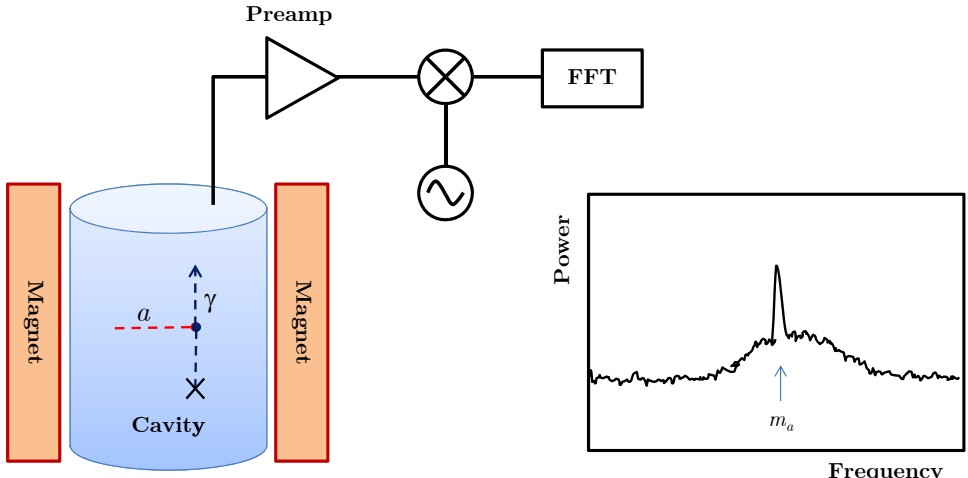

Figure 5: Conceptual arrangement of an axion haloscope. If $m_a$ is within $1/Q$ of the resonant frequency of the cavity, the axion will show as a narrow peak in the power spectrum extracted from the cavity.

distribution truncated at the galactic escape velocity ($\sim$600 km/s). A more recent estimation from N-body simulations provides a more precise shape to the velocity distribution:

$$f(v) \propto \left(v^2\right)^\gamma \exp\left(\frac{v^2}{2\sigma_v^2}\right)^\beta, \tag{31}$$

with $\gamma$, $\beta$ and $\sigma_v$ being fitting constants given in [129]. In any case, the typical dispersion velocity is around $\sigma_v \sim 10^{-3}$. This can be considered an upper limit on the velocity dispersion of DM particles, but particular models may predict finer phase-space substructure. Perhaps the most extreme case is the infall self-similar model developed by Sikivie and collaborators [35, 36]. This model predicts that a substantial fraction of the DM axions in the form of a few velocity streams with much lower values of $\sigma_v$.

In any case, this means that the population of DM axions is better described collectively by a coherent classical field (rather than a "gas" of particles, like the case of WIMPs). The field is coherent over lengths approximately equal to the de Broglie wavelength:

$$\lambda_c \lesssim \frac{\pi/2}{m_a \sigma_v} \sim 200\left(\frac{m_a}{10\,\mu eV}\right)^{-1} \text{m}, \tag{32}$$

which for typical axion masses is well beyond the size of the experiments (and even larger coherence lengths are expected for models with low dispersion streams). The field can then be considered spatially constant in the local region of our experiment and oscillating with a well defined frequency close to the axion mass $\nu_a \sim m_a$. The spread in frequency $\delta\nu_a$ around this value reflects the above-mentioned velocity dispersion, and corresponds to $\delta\nu_a/\nu_a = 10^{-6}$. Its inverse is the axion quality factor $Q_a \sim 10^6$ (once more, for models with low dispersion streams, this peak in frequency is expected to have substructure with much lower $\delta\nu_a$). This is a very important feature for experiments as it will allow to exploit coherent techniques to enhance the signal strength at detection.

### 7.1 Conventional haloscopes

The conventional axion haloscope technique [112] involves a high quality factor $Q$ microwave cavity inside a magnet, where $Q$ can be of order $10^5$. By virtue of the Primakoff conversion, DM axions produce photons in the magnetic field. If the resonant frequency of the cavity matches that of the axion field, the conversion is enhanced by a factor $Q$, and the resulting photons appear as an excited mode of the cavity. This power can then be extracted from the cavity via a suitable port connected to a radio-frequency (RF) detection chain with a low-noise amplifier. The power $P_s$ of such a signal can be calculated to be:

$$P_s = \kappa \frac{Q}{m_a} g_{a\gamma}^2 B^2 |\mathcal{G}_m|^2 V \rho_a \,, \tag{33}$$

where $\kappa$ is the coupling of the cavity to the port, $B$ the external magnetic field, $V$ the volume of the cavity, and $\mathcal{G}_m$ a geometric factor accounting for the mode overlap between the given cavity mode electric field $\mathbf{E}$ and the external magnetic field $\mathbf{B}$:

$$|\mathcal{G}_m|^2 = \frac{\left( \int dV \mathbf{E} \cdot \mathbf{B} \right)^2}{V |\mathbf{B}|^2 \int dV \epsilon \mathbf{E}^2} \,, \tag{34}$$

where the integral is over the entire volume $V$ of the cavity. The modes with higher $\mathcal{G}_m$ are the ones whose electric field is better aligned with the external magnetic field. For example, for a cylindrical cavity and a $\mathbf{B}$ field along the cylindrical axis, the $TM_{0n0}$ modes are the ones that couple with the axion, and the fundamental $TM_{010}$ mode provides the larger geometric factor $|\mathcal{G}_{TM_{010}}|^2 \sim 0.69$.

The signal in Eq. (33) is only valid if the axion frequency matches the resonant frequency of the cavity within the very narrow cavity bandwidth $\sim m_a/Q$. Given that $m_a$ is not known, in order to scan a meaningful range of axion masses the cavity must be tunable in frequency, something that is normally achieved by the implementation of precisely movable pieces that change the geometry of the cavity (e.g. movable rods). The experimental protocol involves a scanning procedure that devotes a small exposure time in each of the frequency points, then moving to the next one, and so forth. Covering a wide mass range poses an experimental challenge.

Figure 5 shows a sketch of the concept of the axion haloscope. A putative signal would appear as a narrow peak at the frequency corresponding to $m_a$ and with an intensity corresponding to Eq. (33). The capability of seeing such a signal will depend also on the level of noise and the exposure time. In absence of systematic effects, the longer the integration time, the smaller the noise fluctuations and the higher the signal-to-noise ratio. In general, the figure of merit $F_{halo}$ of an axion haloscope can be defined as proportional to the time needed to scan a fixed mass range down to a given signal-to-noise ratio. This shows the main parameter dependencies:

$$F_{halo} \propto \rho_a^2 g_{a\gamma}^4 m_a^2 B^4 V^2 T_{sys}^{-2} |\mathcal{G}|^4 Q \,, \tag{35}$$

where $T_{sys}$ is the effective noise temperature of the detector. Typically the noise in these detectors come from thermal photons and therefore it is driven by the physical temperature of the system $T_{phys}$. In reality $T_{sys}$ includes additional components due to e.g. amplifier noise, $T_{sys} = T_{phys} + T_{amp}$. Eq. (35) is useful to see the relative importance of each of the experimental parameters. Note the dependency with $\sim Q$ (instead of $Q^2$ naively expected from Eq. 33), which is due to the fact that improving $Q$ increases the signal strength but reduces the bandwidth of a single frequency point, increasing the total number of steps needed to scan a given mass range. Note also that improvement in $Q$ contributes to $F_{halo}$ only as long as $Q < Q_a$, that is, the axion peak must be contained in the cavity bandwidth, otherwise some

signal will be lost. The improvement shown by Eq. (35) with lower $T_{\text{sys}}$ has also a limit imposed by the presence of vacuum quantum fluctuations, known as the Standard Quantum Limit (SQL) [130,131]. The temperature at which this is relevant is $T_{\text{SQL}} \sim 10(m_a/1\,\mu\text{eV})$ mK. There are ways being currently developed to circumvent this limit, as will be commented below, by using squeezed photon states [132] or single-photon detection [133][12].

Experimental efforts to implement the axion haloscope concept have been led for many years by the ADMX collaboration, which has pioneered many relevant technologies (high Q-cavities inside magnetic fields, RF detection close to the quantum limit and others). The main ADMX setup includes a 60 cm diameter, 1 m long cavity inside a solenoidal $\sim$8 T magnet. The cavity can be tuned in frequency by the precise movement of some dielectric or metallic rods. With this setup, ADMX has achieved sensitivity to axion models in the $\mu$eV range [136] and is currently taking data to extend this initial result at lower temperatures (and thus lower noise). The collaboration has released results [137,138] with sensitivity down to pessimistically coupled axions in the 2.66–3.31 $\mu$eV range (see figure 6), and very recently expanded this range up to 4.2 $\mu$eV with sensitivity at roughly half the DFSZ coupling [139].

In recent years, a number of new experimental efforts are appearing, some of them implementing variations of the haloscope concept, or altogether novel detection concepts, making this subfield one of the most rapidly changing in the axion experimental landscape. Figure 6 shows the current situation, with a number of new players accompanying ADMX in the quest to cover different axion mass ranges. Applying the haloscope technique to frequencies considerably higher or lower than the one ADMX is targeting is challenging, for different reasons. Lower frequencies imply proportionally larger cavity volumes and thus bigger, more expensive, magnets, but otherwise they are technically feasible. The use of large existing (or future) magnets has been proposed in this regard (e.g. the KLASH [140] proposal at LNF, ACTION [141] in Korea, or a possible haloscope setup in the future (Baby)IAXO helioscopes [142]).

Higher frequencies imply lower volumes and correspondingly lower signals and sensitivity. This could be in part compensated by enhancing other experimental parameters (more intense magnetic fields, higher quality factors, noise reduction at detection, etc.). The HAYSTAC experiment [143] at Yale, born in part out of developments initiated inside the ADMX collaboration [144] has implemented a scaled down ADMX-like setup, and has been the first experiment proving sensitivity to QCD models in the decade above ADMX, in particular in the mass range 23.15 - 24.0 $\mu$eV [145]. More recently, HAYSTAC has reported the first axion DM search with squeezed photon states [146], which effectively allows to push the noise limits below the quantum limit [132], reaching sensitivity almost to the KSVZ coupling in the 16.96–17.28 $\mu$eV range. HAYSTAC is also pioneering analysis methodology [147,148] in these types of searches. Another similar program is CULTASK, the flagship project of the recently created Center for Axion and Precision Physics (CAPP) in Korea [149]. CAPP also hosts several other projects and R&D lines, with the general long-term goal of exploring DM axions in the mass range 4-40 $\mu$eV. The first result from this line is the CAPP-PACE pilot experiment [150,151], that has recently produced an exclusion in the 10.1–11.37 $\mu$eV range. A more recent result, from a larger setup CAPP-8TB [152] has produced another excluded region in the 6.62–6.82 $\mu$eV mass range with sensitivity down to the upper part of the QCD band. The QUAX-$a\gamma$ experiment in Frascati has also recently reported [153] a first result with a cavity resonating at an axion mass of 43 $\mu$eV, and read out with a Josephson parametric amplifier whose noise fluctuations are at the SQL, making this the axion haloscope having reached sensitivity to the QCD axion at the highest mass point. Even higher frequencies are targeted by the ORGAN [154] program, recently started in the University of Western Australia. A first pathfinder run has al-

---

[12]It is worth to note in this respect the pioneering, but discontinued, R&D of the CARRACK experiment with Rydberg atoms long ago [134]. Much more recently, the possibility of detecting single photons at these frequencies may come from the progress in superconducting qubits [135].

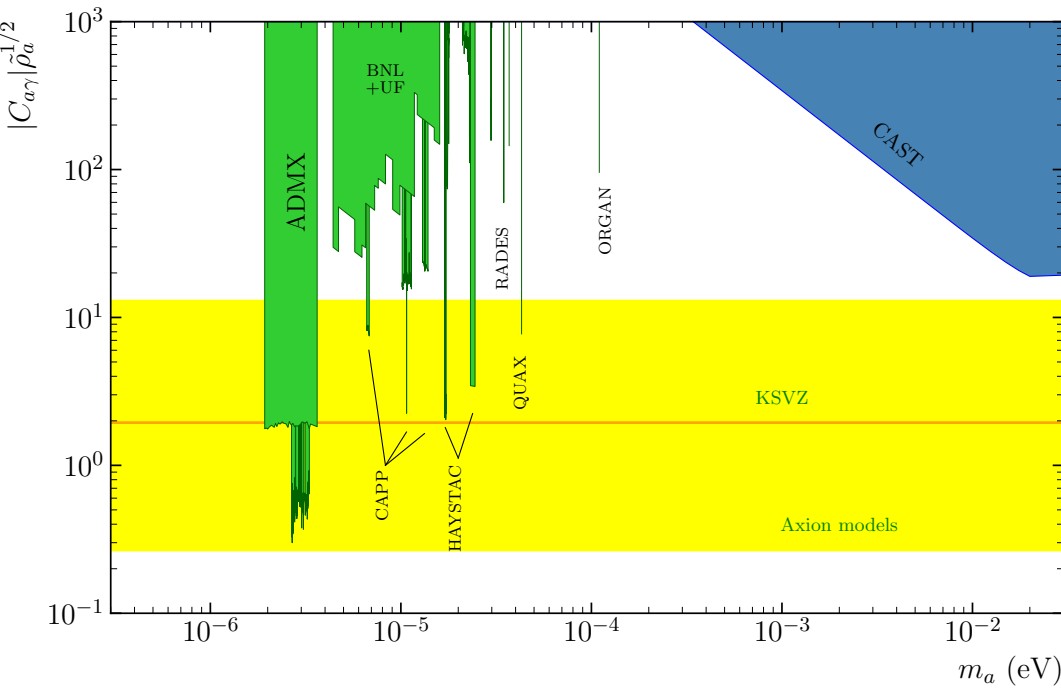

Figure 6: Zoom-in of the region of parameters where most axion dark matter experiments are active (in green). The $y$-axis shows the adimensional coupling $C_{a\gamma} \propto g_{a\gamma}/m_a$ (scaled with the local axion DM density relative to the total DM density, $\tilde{\rho}_a = \rho_a/\rho_{DM}$, to stress that these experiments produce bounds that are dependent on the assumed fraction of DM in the form of axions). Thus the yellow region, where the conventional QCD axion models are, appears now as a horizontal band, but is the same yellow band shown in the other plots of this review.

ready taken place [154], at a fixed frequency of 26.531 GHz, corresponding to $m_a = 110\,\mu$eV. Several groups explore the possibility to increase the cavity $Q$ by coating the inside of the cavity using a superconducting layer. In particular, this strategy has been implemented by the QUAX collaboration, proving an improvement of a factor of 4 with respect to a copper cavity and has performed a single-mass axion search at about $\sim 37\,\mu$eV [155]. Another strategy to reach higher frequencies is to select a higher order mode of the cavity as the one to couple with the axion field, albeit with a lower geometric factor. This has been done by the ADMX "Sidecar" setup, a testbed experiment living inside of and operating in tandem with the main ADMX experiment [156].

Higher frequencies eventually require to increase the instrumented volume, either by combining many similar phase-matched cavities, or by implementing more complex extended resonant structures that effectively decouple the detection volume $V$ from the resonant frequency. The former has already been done long ago for four cavities within the ADMX R&D [157], but going to a much larger number of cavities has been considered not feasible in practice. More recently the CAST-CAPP project [158] is operating several long-aspect-ratio rectangular (i.e. waveguide-like) cavities inserted in CAST dipole magnet at CERN. The option of sub-dividing the resonant cavity is investigated by the RADES project [159], also implemented in the CAST magnet. RADES is exploring the use of arrays of many small rectangular cavities connected by irises, carefully designed to maximally couple to the axion field for a given resonant mode. Data with a 5-subcavity prototype [160] has been used to extract a limit at a fixed axion mass of 34.67 $\mu$eV [161], and more recently a 30-subcavity model is in operation. A similar concept, better adapted to a solenoidal magnet, is being followed at CAPP, with the concept of

a sliced-as-a-pizza cavity [162], which consists on dividing the cylindrical cavity in sections connected by a longitudinal iris along the cylinder's axis of symmetry. A first version with two sub-cavities has been recently used [163] to perform a search in the 13.0–13.9 $\mu$eV mass range. Finally, resonance to higher frequencies with a relatively large resonator can also be achieved by filling it with individually adjustable current carrying wire planes. R&D is ongoing in this direction by the ORPHEUS experiment [164].

## 7.2 Dish antennas and dielectric haloscopes

Going to even higher frequencies requires altogether different detection concepts. Most relevant is the concept of the *magnetized dish antenna* and its evolution, the *dielectric haloscope*. A dielectric interface (e.g. a mirror, or the surface of a dielectric slab) immersed in a magnetic field parallel to the surface should emit electromagnetic radiation perpendicular to its surface, due to the presence of the dark matter axion field [165]. This tiny signal can be made detectable if the emission of a large surface is made to concentrate in a small point, like e.g. in the case of the surface having a spherical shape. This technique has the advantage of being broad-band, with sensitivity to all axion masses at once [13]. This technique is being followed by the BRASS [166] collaboration at U. of Hamburg, as well as G-LEAD at CEA/Saclay [167].

Given that no resonance is involved in this scheme, very large areas are needed to obtain competitive sensitivities. Dielectric haloscopes are an evolution of this concept, in which several dielectric slabs are stacked together inside a magnetic field and placed in front of a metallic mirror. This increases the number of emitting surfaces and, in addition, constructive interference among the different emitted (and reflected) waves can be achieved for a frequency band if the disks are adjusted at precise positions. This effectively amplifies the resulting signal. The MADMAX collaboration [168] plans to implement such a concept, using 80 discs of LaAlO$_3$ with 1 m$^2$ area in a 10 T B-field, leading to a boost in power emitted by the system of a $> 10^4$ with respect to a single metallic mirror in a relatively broad frequency band of 50 MHz. By adjusting the spacing between the discs the frequency range in which the boost occurs can be adjusted, with the goal of scanning an axion mass range between 40 and 400 $\mu$eV (see figure 1). The experiment is expected to be sited at DESY. A first smaller-scale demonstrating prototype will be operated in the MORPURGO magnet at CERN in the coming years, before jumping to the full size experiment. Finally, let us mention that an implementation of the dielectric haloscope concept but at even higher frequencies has been discussed in the literature, with potential sensitivity to 0.2 eV axions and above [169].

## 7.3 DM Radios

For much lower axion masses (well below $\mu$eV), it may be more effective to attempt the detection of the tiny oscillating *B*-field associated with the axion dark matter field in an external constant magnetic field, by means of a carefully placed pick-up coil inside a large magnet [170–172]. Resonance amplification can be achieved externally by an *LC*-circuit, which makes tuning in principle easier than in conventional haloscopes. A broad-band non-resonant mode of operation is also possible [172]. Several teams are studying implementations of this concept [172, 173]. Two of them, the ABRACADABRA [174, 175] and SHAFT [176] experiments, have recently released results with small table-top demonstrators, reaching sensitivities similar to the CAST bound for masses in the $10^{-11}-10^{-8}$ eV range. Another similar implementation, that of BEAST [177], has obtained better sensitivities in a narrower mass range around $10^{-11}$ eV, although its principle has been doubted by the community [178, 179]. Similarly, the more recent result from the ADMX SLIC pilot experiment has probed a few narrow regions around $2 \times 10^{-7}$ eV and down to $\sim 10^{-12}$ GeV$^{-1}$ [180]. Finally, the BASE experiment, whose

---

[13]In practice this is limited by the bandwidth of the photon sensor being used.

main goal is the study of antimatter at CERN, has recently released a result adapting its setup to the search of axions following this concept [181]. In general, this technique could reach sensitivity down to the QCD axion for masses $m_a \lesssim 10^{-6}$ eV, if implemented in magnet volumes of few m$^3$ volumes and a few T fields.

## 7.4 Other techniques

A recent proposal to detect axion DM at even higher mass values involves the use of certain antiferromagnetic topological insulators [182,183]. Such materials contain axion quasiparticles (AQs), that are longitudinal antiferromagnetic spin fluctuations. These AQs have similar dynamics to the axion field, including a mass mixing with the electric field in the presence of magnetic fields. The dispersion relation and boundary conditions permit resonant conversion of axion DM into THz photons in a way that is independent of the resonant frequency. An advantage of this method is the tunability of the resonance with applied magnetic field. The technique could be competitive in the search for DM axions of masses in the 1 to 10 meV range.

Another recently proposed strategy are the "plasma haloscopes", in which the resonant conversion is achieved by matching the axion mass to a plasma frequency. The advantage of this approach is that the plasma frequency is unrelated to the physical size of the device, allowing large conversion volumes. A concrete proposal using wire metamaterials as the plasma, with the plasma frequency tuned by varying the interwire spacing, points to potentially competitive sensitivity for axion masses at $35-400$ eV [184].

At the very low masses, DM axions can produce an oscillation of the optical linear polarization of a laser beam in a bow-tie cavity. The DANCE experiment has already provided proof-of-concept results [185] with a table-top setup, while large potential for improvement exists in scale-up projections.

The techniques mentioned above are all based on the axion-photon coupling. If the axion has relevant fermionic couplings, the axion DM field would couple with nuclear spins like a fictitious magnetic field and produce the precession of nuclear spins. Moreover, by virtue of the same Peccei-Quinn term that solves the strong CP problem, the DM axion field should induce oscillating electric-dipole-moments (EDM) in the nuclei. Both effects can be searched for by nuclear magnetic resonance (NMR) methods. The CASPEr project [186, 187] is exploring several NMR-based implementations to search for axion DM along these directions. The prospects of the technique may reach relevant QCD models for very low axion masses ($\lesssim 10^{-8}$eV). A conceptually similar concept is done by the QUAX experiment, but invoking the electron coupling using magnetic materials [188]. In this case, the sample is inserted in a resonant cavity and the spin-precession resonance hybridises with the electromagnetic mode of the cavity. The experiment focuses on a particular axion mass $m_a \sim 200~\mu$eV, but sensitivity to QCD models will require lowering the detection noise below the quantum limit. The recent experiment NASDUCK [189] has reported competitive limits on $g_{ap}$ and $g_{an}$ from ALP DM interacting with atomic spins, using a quantum detector based on spin-polarized xenon gas. Another technique recently proposed is to search for the axion/ALP induced EDM in the future proton storage ring develop to measure the static proton EDM [190].

DM axions can produce atomic excitations in a target material to levels with an energy difference equal to the axion mass. This can again happen via the axion interactions to the nuclezi or electron spins. The use of the Zeeman effect has been proposed [191] to split the ground state of atoms to effectively create atomic transition of energy levels that are tunable to the axion mass, by changing the external magnetic field. The AXIOMA [192, 193] project has started feasibility studies to experimentally implement this detection concept. Sensitivity to axion models (with fermion couplings) in the ballpark of $10^{-4}-10^{-3}$ eV could eventually be achieved if target materials of $\sim$kg mass are instrumented and cooled down to mK temperatures. For a more thorough review of the possibilities that atomic physics offer to axion

physics we refer to section 1.4 of Ref. [5].

Before concluding this section, let us mention that a DM axion with keV mass (or higher) and with sufficiently strong coupling to electrons would show up in low background massive detectors developed for WIMP searches [194–196], as a non-identified peak at an energy equal to the mass, by virtue of the axioelectric effect. The recent XENON1T low-energy electronic recoil event excess [197] could be interpreted as such a signal.

# 8 Solar axion experiments

If axions exist, they would be produced in large quantities in the solar interior. The most important channel are Primakoff solar axions. They are a robust prediction by virtually any axion model, only requiring a non-zero $g_{a\gamma}$ and relying on well-known solar physics. Axions coupled to electrons offer additional production channels. Once produced, axions get out of the Sun unimpeded and travel to the Earth, offering a great opportunity for direct detection in terrestrial experiments. The leading technique to detect solar axions are axion helioscopes [112], one of the oldest concepts used to search for axions. Axion helioscopes (see Figure 7) are sensitive to a given $g_{a\gamma}$ in a very wide mass range, and after several past generations of helioscopes, the experimental efforts are now directed to increase the scale and thus push sensitivity to lower $g_{a\gamma}$ values. Contrary to the scenario described in the haloscope frontier, with a plethora of relatively small, sometimes table-top, experiments, most of the helioscope community has coalesced into a single collaboration, IAXO, to face the challenges to build a large scale next-generation helioscope. Indeed, the IAXO collaboration is by far the largest experimental collaboration in axion physics, currently with about 125 scientists from 25 different institutions.

## 8.1 Solar axions

Photons from the solar plasma would convert into axions in the Coulomb fields of charged particles via the Primakoff axion-photon conversion. The produced axions have energies reflecting the typical solar core photon energies, i.e. around $\sim 3$ keV. Therefore they are relativistic and the predicted flux is independent on $m_a$ (as long as $m_a \lesssim$ keV, which is the case for the QCD axion). A useful analytic approximation to the differential flux of Primakoff solar axions at Earth, accurate to less than 1% in the 1–11 keV range, is given by [198]:

$$\frac{\mathrm{d}\Phi_a}{\mathrm{d}E} = 6.02 \times 10^{10} \left(\frac{g_{a\gamma}}{10^{-10}\mathrm{GeV}^{-1}}\right)^2 E^{2.481} e^{-E/1.205} \frac{1}{\mathrm{cm}^2 \text{ s keV}} \,, \tag{36}$$

where $E$ is the axion energy expressed in keV. This Primakoff spectrum is shown in Fig. 8 (left). As seen, it peaks at $\sim 3$ keV and exponentially decreases for higher energies. Once the existence of a non-zero $g_{a\gamma}$ is assumed, the prediction of this axion flux is very robust, as the solar interior is well-known. A recent study of the uncertainties [199] confirms a statistical uncertainty at the percent level, although the number of axions emitted in helioseismological solar models is systematically larger by about 5% compared to photospheric models. At energies below $\sim$keV the uncertainties are larger as other processes can contribute. Recent works have studied other interesting solar axion production channels that have not been exploited experimentally yet. On one side, axions can also be produced in the large scale magnetic field of the Sun. In particular, longitudinal or transversal plasmons can resonantly convert, leading to different detectable populations at sub-keV energies, with a dependence on the particular magnetic field profile of the Sun (and, for the case of transverse plasmons, on the ALP mass) [200–202].

In non-hadronic models axions couple with electrons at tree level. This coupling allows for additional mechanisms of axion production in the Sun [82]: the ABC axions already introduced

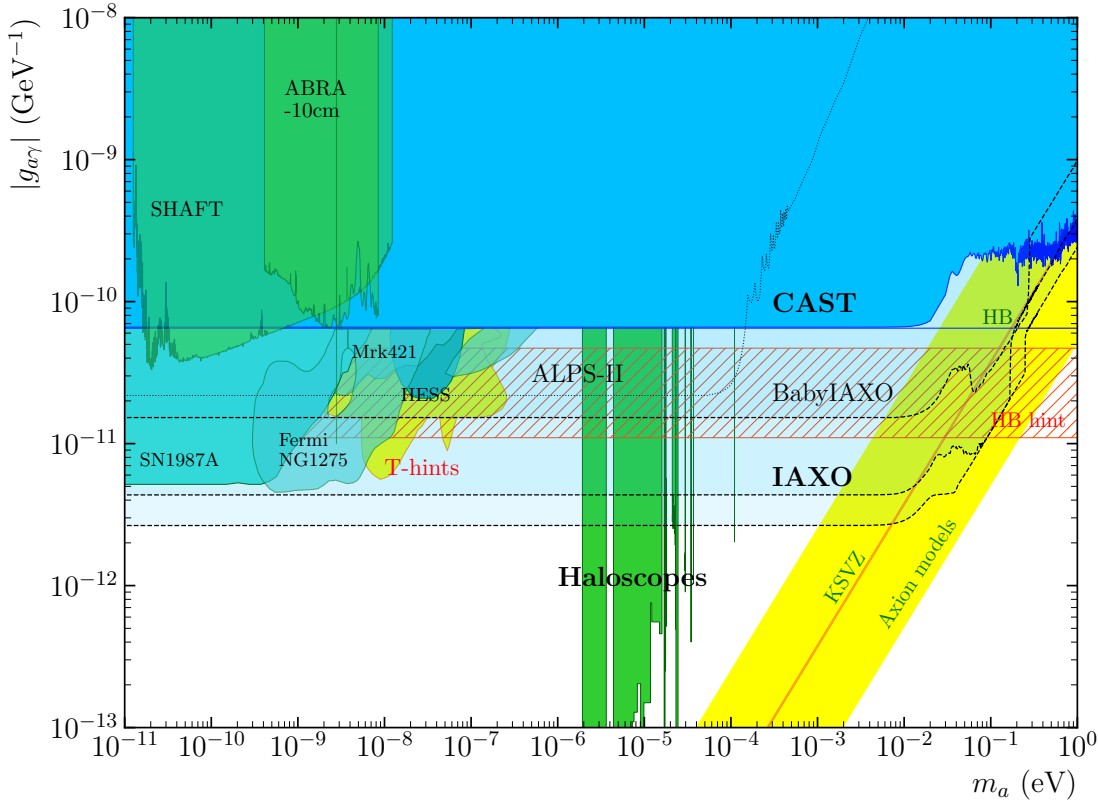

Figure 7: Excluded regions and sensitivity prospects in the ($g_{a\gamma}$, $m_a$) plane, with a focus in the $g_{a\gamma}$ range relevant for helioscopes. Most relevant is the area excluded by CAST, as well as prospects from future helioscopes like BabyIAXO and IAXO (for the latter two scenarios, nominal and enhanced, or IAXO+, are considered), and the LSW experiment (dashed and dotted lines). The "transparency hint" regions commented in section 4.2 are the yellow regions at low masses labelled as "T-hint". The horizontal branch (HB) hint explained in section 4.1.1 is indicated as the red-dashed band labelled "HB hint". All other green and blue regions are exclusions from the different experiments and considerations explained in the text.

in section 2.7. Figure 3 shows the Feynman diagrams of all these processes, namely, atomic axio-recombination and axion-deexcitation, axio-Bremsstrahlung in electron-ion or electron-electron collisions and Compton scattering with emission of an axion.

The spectral distribution of ABC solar axions is shown on the right of Figure 8. Although the relative strength of ABC and Primakoff fluxes depends on the particular values of the $g_{ae}$ and $g_{a\gamma}$ couplings, and therefore on the details of the axion model being considered, for non-hadronic models the ABC flux tends to dominate. Although all processes contribute substantially, free-free processes (bremsstrahlung) constitute the most important component, and are responsible for the fact that ABC axions are of somewhat lower energies than Primakoff axions, with a spectral maximum around ~1 keV. This is because the axio-bremsstrahlung cross-section increases for lower energies and, in the hot solar core, electrons are more abundant than photons, and their energies are high with respect to atomic orbitals. In addition, the axio-deexcitation process is responsible for the presence of several narrow peaks, each one associated with different atomic transitions of the species present in the solar core. These two features would be of crucial importance in the case of a positive detection to confirm an axion discovery.

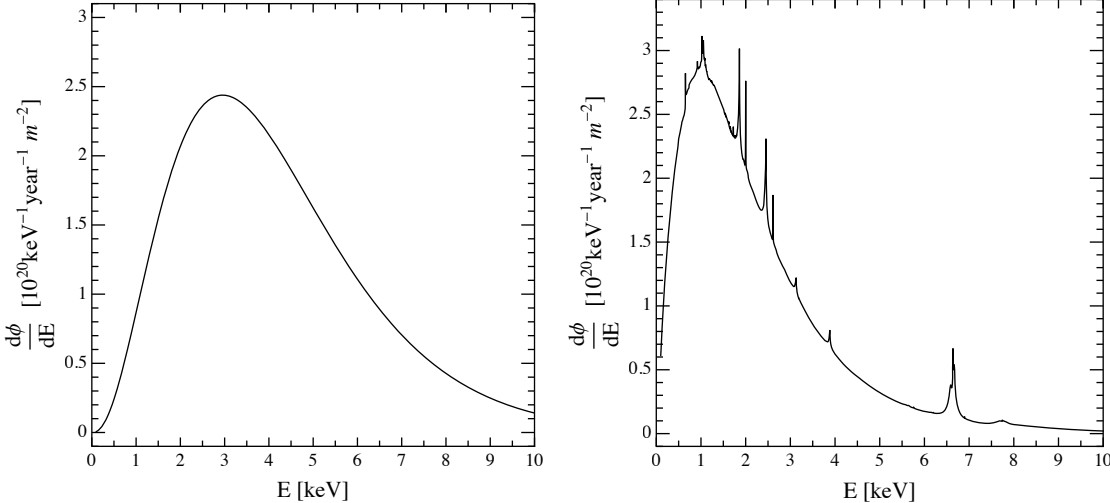

Figure 8: Solar axion flux spectra at Earth by different production mechanisms. On the left, the most generic situation in which only the Primakoff conversion of plasma photons into axions is assumed. On the right the spectrum originating from the ABC processes [82, 203]. The illustrative values of the coupling constants chosen are $g_{a\gamma} = 10^{-12}$ GeV$^{-1}$ and $g_{ae} = 10^{-13}$. Plots from [204].

In spite of the above, due to the fact that $g_{ae}$ is more strongly bounded from astrophysical considerations than $g_{a\gamma}$ (see section 4) the sensitivity of experiments to ABC axions has not been sufficient so far to reach and study unconstrained values of $g_{ae}$. This may change with the next generation of solar axion helioscopes, like IAXO, that will enjoy sensitivity to values down to $g_{ae} \sim 10^{-13}$, as will be commented in section 8.2.

For the sake of completeness, we should mention that the existence of axion-nucleon couplings $g_{aN}$ also allows for additional mechanisms of axion production in the Sun. These emissions are mono-energetic and are associated with particular nuclear reactions in the solar core. Some examples of the emissions that have been searched for experimentally are: 14.4 keV axions emitted in the M1 transition of Fe-57 nuclei, MeV axions from $^7$Li and D$(p, \gamma)^3$He nuclear transitions or Tm$^{169}$ (see [2] for details and references).

## 8.2 Axion helioscopes

The axion helioscope detection concept [112] invokes the conversion of the solar axions back to photons in a strong laboratory magnet. The resulting photons keep the same energy as the incoming axions, and therefore they are X-rays that can be detected in the opposite side of the magnet when it is pointing to the Sun (see Fig. 9). The conversion process inside the heliscope's magnet is conceptually similar to the one presented above for LSW experiments. The probability of conversion in a magnet of constant transverse magnetic field $B$ and length $L$ can be expressed as [112, 198, 205]:

$$\mathcal{P}(a \to \gamma) = 2.6 \times 10^{-17} \left( \frac{g_{a\gamma}}{10^{-10} \text{ GeV}^{-1}} \right)^2 \left( \frac{B}{10 \text{ T}} \right)^2 \left( \frac{L}{10 \text{ m}} \right)^2 \mathcal{F}(qL), \qquad (37)$$

where $\mathcal{F}(qL)$ is a form factor to account for the loss of coherence:

$$\mathcal{F} = \left( \frac{2}{qL} \right)^2 \sin^2 \left( \frac{qL}{2} \right), \qquad (38)$$

with $q = m_a^2/2E_a$ being the momentum transfer (or momentum difference between the photon and axion waves) and $E_a$ the energy of the incoming axion. As in the LSW case, if $qL \ll 1$, $\mathcal{F}(qL) \to 1$ but otherwise $\mathcal{F}(qL)$ starts decreasing and so does the probability of conversion. For solar axion energies, and typical helioscope magnet lengths ($\sim 10$ m) this happens for axion masses around 0.01 eV. Therefore, for $m_a < 0.01$ eV, the sensitivity of an axion helioscope is flat in $m_a$, as can be seen in Figure 7.

Figure 9 shows the typical configuration of axion helioscopes. Due to the dependencies expressed in Eq. (37), dipole-like layouts for the magnet are preferred, that is, relatively long (in the Sun's direction) with a magnetic field in the transverse direction. The magnet is placed on a moving platform that allows to point it to the Sun and track it for long periods. At the end of the magnet opposite to the Sun, the detection line(s) are placed. In modern optically-enhanced versions of helioscopes, X-ray optics are placed just at the end of the magnet bore to focus the almost-parallel beam of photons from axion-conversion into small focal spots. This allows to use relatively large magnet transverse areas, keeping a relatively small detector, and thus increasing the signal-to-noise ratio. X-ray optics are built following techniques developed for X-ray astronomy missions, based on the high reflectivity of X-rays when impinging a mirror with small grazing angle. These optics look like a collection of conical mirrors, one nested inside the next one, until covering the whole magnet area. The X-ray detectors are then placed at the focal points of the optics, and they need to be only slightly larger than the focal spot size ($\sim$cm$^2$). The presence of solar axions will manifest itself as an excess of counts over the detector background, the latter measured in the detector area outside the signal spot, or during periods in which the magnet is not pointing to the Sun. The detector should be energy-resolving and pixelated, so that the energy distribution of the detected photons, as well as their spatial distribution on the detector plane (the signal "image") can be compared with expectations in case of a positive signal (the latter should correspond to the angular distribution of solar axion emission spatial distribution convoluted with the optics response, or "point spread function"). Because the background is measured and statistically subtracted from the "signal data", the signal-to-noise ratio in axion helioscopes goes with the background fluctutations rather that the background itself ($\sqrt{n}$ versus $n$). In general the figure of merit of an axion helioscope $F_{\text{helio}}$ can be defined as proportional to the signal to noise ratio for a given value of $g_{a\gamma}$, so that:

$$F_{\text{helio}} \propto B^2 L^2 \mathcal{A} \, \frac{\epsilon_d \epsilon_o}{\sqrt{ba}} \, \sqrt{\epsilon_t t} \,, \tag{39}$$

where $B$, $L$ and $\mathcal{A}$ are the transverse magnetic field, length and cross-sectional area of the magnet respectively, $\epsilon_o$ is the throughput of the optics (or focalization efficiency), $a$ the signal spot size after focalization, $\epsilon_d$ the detection efficiency, $b$ the normalized (in area and time) background of the detector, $\epsilon_t$ is the data-taking efficiency, i. e. the fraction of time the magnet tracks the Sun (a parameter that depends on the extent of the platform movements) and $t$ the duration of the data taking campaign.

So far we have assumed the magnet bores are in vacuum. This is what is called baseline (or phase-I) configuration. In order to attain sensitivity for axion masses above the value above which $\mathcal{F}(qL)$ drops due to lack of coherence (i.e. $m_a \gtrsim 0.01$ eV) the bores can be filled with a buffer gas [206]. This gas provides the photon with a mass and restores the coherence for a narrow window of axion masses around the photon refractive mass. In this gas phase (or phase-II) of the experiment the pressure of the gas is changed in steps and the data taking follows a scanning procedure in which the experiment is sensitive to different small mass interval in each step (similar to axion haloscopes, only this time the relative width of the step in mass is of the order $\mathcal{O}(10^{-2})$). Note that the buffer gases can also be used in LSW experiments (see e.g. [207]), although is in helioscopes where it can make a difference in the sensitivity of the experiments, allowing to access QCD axion models at high masses.

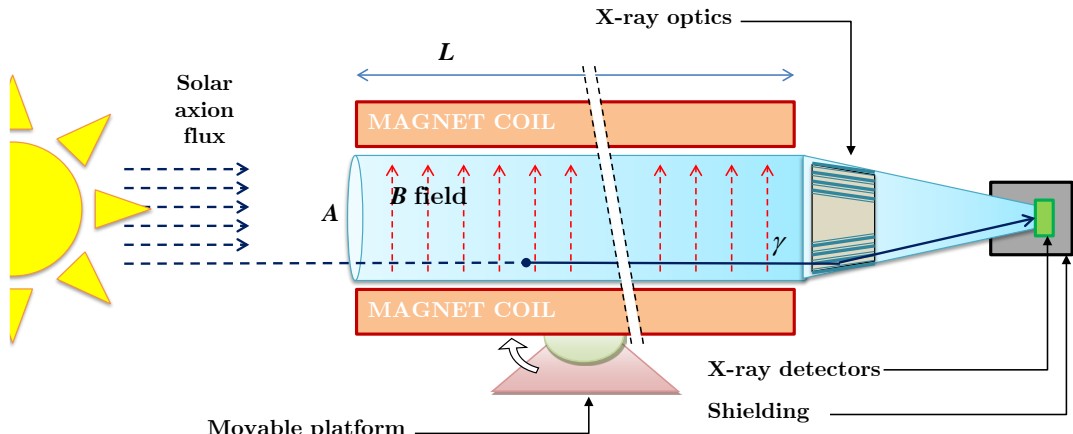

Figure 9: Conceptual arrangement of an enhancged axion helioscope with X-ray focussing. Solar axions are converted into photons by the transverse magnetic field inside the bore of a powerful magnet. The resulting quasi-parallel beam of photons of cross sectional area *A* is concentrated by an appropriate X-ray optics onto a small spot area *a* in a low background detector. Figure taken from [212].

The strategy described above has been followed by the CERN Axion Solar Telescope (CAST) experiment, using a decommissioned LHC test magnet that provides a 9 T field inside the two 10 m long, 5 cm diameter magnet bores. CAST has been active for more than 15 years at CERN, going through several data taking campaigns, and represents the state-of-the-art in the search for solar axions. It has been the first axion helioscope using X-ray optics. The latest solar axion result [208][14] sets an upper bound on the axion-photon coupling of:

$$g_{a\gamma} < 0.66 \times 10^{-10} \text{ GeV}^{-1}, \tag{40}$$

for $m_a \lesssim 0.01$ eV. Figure 7 shows the full exclusion line. The wiggly extension at higher masses, up to about 1 eV is the result of the scanning with a buffer gas in the bores [209–211], which has allowed CAST to actually probe the QCD band in those masses. The limit (40) competes with the strongest bound coming from astrophysics. Advancing beyond this bound to lower $g_{a\gamma}$ values is now highly motivated [212], not only because it would mean to venture into regions of parameter space allowed by astrophysics, but also because the astrophysical hints mentioned in section 4 seem to point at precisely this range of parameters. CAST has also searched for solar axions produced via the axion-electron coupling [203] (and axion-nucleon in [213, 214]) although the very stringent astrophysical bound on this coupling remains so far unchallenged by experiments.

The successor of CAST is the International Axion Observatory (IAXO) [215], a new generation axion helioscope, aiming at the detection of solar axions with sensitivities to $g_{a\gamma}$ down to a few $10^{-12}$ GeV$^{-1}$, a factor of 20 better than the current best limit from CAST (a factor of more than $10^4$ in signal-to-noise ratio). This leap forward in sensitivity is achieved by the realization of a large-scale magnet, as well as by extensive use of X-ray focusing optics and low background detectors.

The main element of IAXO is thus a new dedicated large superconducting magnet [216], designed to maximize the helioscope figure of merit. The IAXO magnet will be a supercon-

---

[14]This result was obtained from data taken in 2014-15, and since then the experiment is hosting different new exploratory setups like the haloscopes described in the previous section.

ducting magnet following a large multi-bore toroidal configuration, to efficiently produce an intense magnetic field over a large volume. The design is inspired by the ATLAS barrel and end-cap toroids, the largest superconducting toroids ever built and presently in operation at CERN. Indeed the experience of CERN in the design, construction and operation of large superconducting magnets is a key aspect of the project.

As already mentioned, X-ray focalization relies on the fact that, at grazing incident angles, it is possible to realize X-ray mirrors with high reflectivity. IAXO envisions newly-built optics similar to those used onboard NASA's NuSTAR satellite mission, but optimized for the energies of the solar axion spectrum. Each of the eight ∼60 cm diameter magnet bores will be equipped with such optics. At the focal plane of each of the optics, IAXO will have low-background X-ray detectors. Several detection technologies are under consideration, but the most developed ones are small gaseous chambers read by pixelised microbulk Micromegas planes [217]. They involve low-background techniques typically developed in underground laboratories, like the use of radiopure detector components, appropriate shielding, and the use of offline discrimination algorithms. Alternative or additional X-ray detection technologies are also considered, like GridPix detectors, Magnetic Metallic Calorimeters, Transition Edge Sensors, or Silicon Drift Detectors. All of them show promising prospects to outperform the baseline Micromegas detectors in aspects like energy threshold or resolution, which are of interest, for example, to search for solar axions via the axion-electron coupling, a process featuring both lower energies than the standard Primakoff ones, and monochromatic peaks in the spectrum.

An intermediate experimental stage called BabyIAXO [218] is the near term goal of the collaboration. BabyIAXO will test magnet, optics and detectors at a technically representative scale for the full IAXO, and, at the same time, it will be operated and will take data as a fully-fledged helioscope experiment, with sensitivity beyond CAST (see Figure 7). It will be located at DESY, and it is expected to be built in 2-3 years.

The expected sensitivity of BabyIAXO and IAXO in the $(g_{a\gamma}, m_a)$ plane is shown in Figure 7, both including also a phase II result at high energies. The IAXO projection includes two lines, one corresponding to nominal expectations and another one a more optimistic projection with a ×10 better $F_{\text{helio}}$. The sensitivity of IAXO to $g_{ae}$ via the search of ABC axions (not shown in the plots) will be for the first time competitive with astrophysical bounds and in particular sufficient to probe a good part of the hinted range from the anomalous cooling of stars. We refer to [100] for more details on this and other the physics potential of BabyIAXO and IAXO.

## 8.3 Other techniques to search for solar axions

A variant of the helioscope technique, dubbed AMELIE [219], can be realized in a magnetized large gaseous detector (e.g. a time projection chamber). In this configuration, the detector gaseous volume plays both the roles of buffer gas where the Primakoff conversion of solar axions takes place, and X-ray detection medium. Contrary to standard helioscopes, in which the resulting X-rays need to cross the buffer gas to reach the detectors, here high photoabsorption in the gas is sought. Therefore, high pressures or high-$Z$ gases are preferred. Due to the short range of the X-rays in the gas, the coherence of the conversion is lost, there is no privileged direction and moving the magnet to track the Sun is no longer necessary. Still the signal depends on the $B$ field component perpendicular to the axion incident direction and therefore even in a stationary magnet a daily modulation of the signal is expected, which give a useful signal signature. The technique could have some window of opportunity at higher masses $\gtrsim 0.1$ eV where buffer gas scanning in helioscopes is increasingly difficult.

Axion-photon conversion (and viceversa) can also happen in the atomic electromagnetic field inside materials. In the case of crystalline media, the periodic structure of the field imposes a Bragg condition, i.e., the conversion is coherently enhanced if the momentum of the incoming particle matches one of the Bragg angles [220]. This concept has been applied to

the search for solar axions with crystalline detectors [221,222]. The continuous variation of the relative incoming direction of the axions with respect to the crystal planes, due to the Earth rotation, produces very characteristic sharp energy- and time-dependent patterns in the expected signal in the detector, which can be used to effectively identify a putative signal over the detector background. This technique has been used as a byproduct of low-background underground detectors developed for WIMP searches [194,196,223–227]. However, in the mass range where helioscopes enjoy full coherent conversion of axions, the prospects of this technique are not competitive [228,229].

Finally, solar axions could also produce visible signals in ionization detectors by virtue of the axioelectric effect [230–234], most relevantly, in large liquid Xe detectors [195,235–237]. However, the sensitivity to $g_{ae}$ is still far from the astrophysical bound. Interactions via nucleon coupling can also be used. For monochromatic solar axions emitted in M1 nuclear transitions, a reverse absorption can be invoked at the detector, provided the detector itself (or a component very close to it) contains the same nuclide, as e.g. in $Fe^{57}$ [238,239], $Li^7$ [240] or $Tm^{169}$ [241]. The upper limits to the nucleon couplings obtained by this method are however larger than the bounds set by astrophysics. As a final comment, a combination of different couplings at emission and detection can also be invoked. The recent XENON1T excess [197], mentioned in a previous section, has also been interpreted as a signal of solar axions via a combination of couplings at emission and detection, including axion-photon conversion in the atomic field of the Xe atoms [242] (this time with no Bragg-like effect). In all cases, the values of the couplings are already excluded by CAST or by astrophysical bounds.

# 9 Conclusions and prospects

Axions and axion-like particles at the low mass frontier appear in very motivated extensions of the SM. For long considered "invisible", very light axions are now at reach of current and near-future technologies in different parts of the viable parameter space. The field is now undergoing a blooming phase. As has been shown in this course, the experimental efforts to search for axions are rapidly growing in intensity and diversity. Novel detection concepts and developments are recently appearing and are being tested in relatively small setups, yielding a plethora of new experimental initiatives. In addition to this, consolidated detection techniques are now facing next-generation experiments with ambitious sensitivity goals and challenges related to large-scale experiments and collaborations. As an example of the importance that this subfield is getting, let us mention that axion searches are explicitly mentioned in the last Update of the European Strategy for Particle Physics. The near and mid-term sensitivity prospects show promise to probe a large fraction of the axion parameter space, and a discovery in the coming years is not excluded. Such a result would be a breakthrough discovery that could reshape the subsequent evolution of Particle Physics, Cosmology and Astrophysics.

# Acknowledgements

I would like to thank the organizers of Les Houches Dark Matter School 2021 for their invitation to impart this course, that has allowed these notes to get written. I would also like to thank my many collaborators in axion physics (too many to get listed here), from whom I have learnt (and I continue to learn) so many things. Particularly pertinent in this case is my gratitude to J. Redondo, with whom I co-authored the recent review [2] that I have cited in several places here and that I have used when preparing this text; and M. Giannotti, for his help with the astrophysics part, both in the text (as I have used the excellent review [3] of which he is a co-

author) as well as with the material he kindly offered for my slides in this part; and similarly to J. Vogel, with whom I co-authored the chapter on solar axions in the "axion textbook" [6] that I mentioned in the introduction and that I believe is an additional (and pedagogical) source of information for the target students of this course. I would also like to thank the students that attended the school, their interest and recognition being the best reward for a lecturer, and in particular to S. Mutzel for her careful reading of these notes. Finally, I would like to acknowledge the funding bodies that are currently supporting directly my research activity: the European Research Council (ERC) under the European Union's Horizon 2020 research and innovation programme, grant agreement ERC-2017-AdG788781 (IAXO+), as well as the Spanish Agencia Estatal de Investigación under grant PID2019-108122GB.

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
