# Peer review of "An introduction to axions and their detection"

_SciPost Physics Lecture Notes, doi:SciPost Phys. Lect. Notes 45 (2022)_

## Round 1 · Referee Report · Anonymous (Referee 1) · 2021-9-19

Strengths

These lecture notes give a comprehensive yet very readable overview of the cutting edge in the field of axions and their detection. The level of detail is appropriate for an introduction and the majority of the relevant literature, including other reviews, is cited.

Weaknesses

The notes are in good shape but are in good need of a proof read, and to fill in some gaps where references are missing (I have provided a list).

I think there is a slight over-weighting towards the discussion of helioscopes. Much more time is spent on the technical details of them than on other aspects (this could be pointed out in the introduction for instance).

Report

I can recommend this manuscript for publication, I only have relatively minor comments and corrections that primarily centre around typos and missing references/discussion.

Requested changes

Global change: the figures do not line up well to the order they appear in the text.

Page 1: * Abstract: "and as dark matter" → "and as a dark matter"

Page 2: * "SN" should be "SM"

Page 3: * "pNB" should be "pNG"

Page 4: * "Viewed the QCD Lagrangian in isolation" → "Viewing the QCD Langrangian in isolation"

Page 5: * 10^7 GeV^-1 → 10^7 GeV * Eq. 3, there is an updated EDM measurement here https://arxiv.org/pdf/2001.11966.pdf

Page 6: * "Are that do not" → "are that they do not"

Page 7: * "issue that will..." → "an issue that will..."

Page 9: * "in additional" → "in addition"

Page 10: * Figure 1: Thee minimum of the potential would usually be at an angle of 0.

Page 11: * Space between "(VR)mechanism" * "production mechanism is each scenario" → in each scenario

Page 12: * Just above Eq. 21, it says Omega^pre rather than Omega^post

Page 13: * Table 1: "up to several times it"? * "dilte them away" → "dilutes them away"

Page 14: * Discussion on TD simulations seems tobe missing references to e.g. https://arxiv.org/abs/1809.09241 https://arxiv.org/abs/1906.00967 https://arxiv.org/abs/2108.05368

Page 15: * N_DM instead of N_DM is used a few times

Page 16: * There was another more recent telescope search done in https://arxiv.org/abs/2009.01310

Page 17: * Miniclusters bullet point: no references anywhere here * "as byproduct" → "as a byproduct"

Page 18: * "Classical" → "Classic" * No CL or significance given fro the 7e-10 GeV^-1 stellar bound

Page 20: * "Whose Feynman diagram" → diagrams * Eq. 25: The more stringent bound is 1.3e-13 taken from omega centauri in the recent Capozzi and Raffelt paper

Page 21: * Eq. 27: This doesn't seem to match Eq.(3.3) of https://arxiv.org/pdf/1906.11844.pdf * No citation given for the NS cooling bounds.

Page 22: * "If the field extend" → extends

Section 4.3: One could also mention here: * DM axions falling into neutron star magnetospheres and producing a radio line https://arxiv.org/abs/2004.00011 https://arxiv.org/abs/2008.11188 https://arxiv.org/abs/2107.01225 * stellar axions converting in MW B-field from star clusters https://arxiv.org/abs/2008.03305 * SN distance ladder measurements https://arxiv.org/abs/2011.05993 * Diffuse background of ALPs https://arxiv.org/abs/2008.11741

Page 23: * Superradiance: There are also lighter masses that are constrained from Supermassive black holes rather than just the stellar mass ones where this bound comes from. See e.g. https://arxiv.org/abs/2009.07206 * Text beginning "The search for low energy axions..." is a repetition of earlier text

Page 24: * "Haloscopes rely on the assumption of the" → "that"

Page 28: * Eq. 32: personally i don’t believe one can use N-body simulations to estimate this. Certainly there are other works using different simulaitons that find different functional forms e.g. https://arxiv.org/abs/1811.11763

Page 30: * "poses a experimental" → an experimental * Soleoidal → Solenoidal

Page 32: * "this type of searches" → "these types of searches"

Page 33: * 10.000 → 10,000 * It should be noted that the principle behind BEAST---namely that you can sense axion-induced electric fields down in the long-wavelength quasi-static regime---has been seriously doubted by the community, see e.g. https://arxiv.org/abs/1809.10709 https://arxiv.org/abs/1812.05487 * "and few T" → "and a few T"

Page 34 * "similar dynamics than" → "similar dynamics to" * nuclear spin expts: could mention the recent NASDUCK experiment here https://arxiv.org/abs/2105.04603 * nucleai → nuclei * "use of Zeeman" → "use of the Zeeman" * "split he" → "split the" * "Sensitiviy" → "Sensitivity" * "in the in the" * XENON1T excess: this is only true when interpreted as a solar axion flux, i think the DM ALP interpretation is not constrained

Page 35 * Last paragraph is repeated text

Page 36: * Doesn't Figure 7 also have IAXO+ shown too? * "Despite of the"

Page 38: * "Allows to use"

Page 41: * "Which give a useful" → gives

---

## Round 1 · Referee Report · Anonymous (Referee 2) · 2021-10-11

Strengths

1) Extremely nice, comprehensive and thorough overview of the field. 2) Includes an exhaustive list of references, which is perfect for beginners in the field, but also very helpful to everybody else.

Weaknesses

1) Due to the nature of the manuscript it is partially of list-like character (in particular in sections 7.1 (end) to 7.4). This part could use a bit more guidance for the reader. 2) The part on IAXO is a bit too detailed/enthusiastic compared to the other discussed experiments.

Report

The manuscript definitely covers a subject of ongoing (rather increasing) interest to the research community and provides a correct, systematic and intelligible presentation of the material. Thus the journal's acceptance criteria are met, actually particularly well.

The manuscript is an extremely nice, systematic, comprehensive and thorough overview of the field of axions and experiments for their detection for beginners to the field. What should be mentioned in particular is the exhaustive list of up-to-date references, which is helpful to beginners and other readers alike.

I would definitely be happy to see these lecture notes published.

Requested changes

1) Chapters 7.1-7.4 (mainly the part on the haloscopes) are a bit of a list-like character. This is not necessarily bad as it is helpful to have a close-to exhaustive list of the relevant experiments. Still, it would be more useful, if there could be something to guide the reader, e.g. an actual list or plot showing the regions which the different experiments probe. 2) In some parts there is references to "later on" or similar - please replace that by the reference to the chapter. 3) The manuscript should be proof-read by an English native speaker. 4)Figures should appear approximately where they are mentioned for the first time. 5) A plot showing the (in particular astrophysical) hints might be helpful. Some of them are included in Figure 7. It might also be sufficient to make clearer in this plot, what are hints, experiments, exclusions. 6) List of minor corrections, typos etc: -page 2: SN==>SM -page 3: pNB==>pNG -page 5: being g==> g being -page 6: first term and second term mixed up, "that the electroweak scale" ==> "than the electroweak scale", "that do not contain" ==> "that they do not contain" -page 7: "issue that will be commented later" -page 8: contraint==>constrain, arise==>arises -page 9: repetition of convention a/A, I would consider once sufficient; constraint==>constrain -page 10: green dot barely visible in plot -page 11: (VR)mechanism==>(VR) mechanism, mechanism is==>mechanism in -page 12: mention the $\Omega_{A,VR}$ and $\Omega_{DM}$ in the text when introducing them. Explain why assume $\theta^2_iF \sim1$ at all. -page 13: refer to table 1 in text, Becasue==>Because -page 14: take==>takes -page 15: introducing anthropic window once is sufficient -page 16: that the age==> than the age -page 17: of the alter the , the drove ==> that drove, Earch==>Earth, chamaleons==>chameleons -page 18: section==>Section -page 19: ref [73]==>[73] -page 20: brehmsstrahlung==>bremsstrahlung -page 21: one example of "see later", please name section, but however==>however -page 22: field==>fields, Earch==>Earth, is magnetic field ==> magnetic field is -page 23: add reference to third bullet point -page 24: another techniques==>another technique; LSW also possible with resonators only in production cavity -page 25: explanation of $\beta_R$? , that is more complex admittedly, but not explained as implied -page 27: ALPs==>ALPS, ALPS-II==>ALPS II, length of ALPS now about 2*125m (see e.g. 2021 PATRAS presentation) -page 30: proportional the ==>proportional to the -page 31: the the==> the -page 32: reference to CULTASK?, proving proven -page 33: haloscopes is ==> are -page 34: explain why containing axion quasiparticles, exist==>exists, similar is done==>; nucleai==>nuclei, in the in the==> in the -page 35: by a virtually==>by virtually; rephrase sentence with axions travelling to Earth, it is clear, what is meant, but as written is not true for all axions -page 36: ALP-II==>ALPS II , explanation of ABC is repetition -page 37: later on==>refer to relevant section, similar that==>similar to -page 38: with q=... being -page 39: use of buffer gas for other experiments? LSW?, x-ray==>X-ray; one bracket missing at the end of first paragraph -page 40: Figure 7 (without bracket) -page 41: bit too enthusiastic, previous section==>mention section

---

## Round 1 · Referee Report · Anonymous (Referee 3) · 2021-10-29

Strengths

  1. The notes presented here are extremely comprehensive and cover all of the relevant topics in the field of axion physics (both from a theoretical and experimental view).

  2. The theory sections are extremely accessible to a non theorist (such as myself) while still including all of the relevant details. This is a perfect balance for an introductory overview.

  3. While the experimental overview is at times sparse on details the author provides a comprehensive list of citations to further reading for those that are interested in learning more about specific searches.

Weaknesses

  1. The grammar needs some work. I found a handful of minor typos and grammar mistakes which didn't impact the physics content but should be corrected.

  2. Sections 6-8 mostly read as lists of experiments but this is understandable as the scope of the note is large, making it difficult to spend too much time on any one topic.

  3. Less a complaint and more a comment but there is a noticeable bias towards helioscopes (specifically IAXO) both in time spent discussing and general tone used in their discussion. I don't think this is an issue, since these are lecture notes, but I do think its worth pointing out there is a slight and noticeable bias.

Report

These notes give a clear and concise overview of the axion field. The overview is well written and covers all of the necessary topics and experiments at a level appropriate for an introductory note.

I would certainly have benefited from these notes when I was first introduced to the field and as such I think these notes should be published once the minor comments (mostly grammar) I have are addressed.

Requested changes

  1. in Section 3.2 you discuss TDs and some simulations to predict the axion mass. I recently saw an update from one of the authors of (arXiv:2108.05368) which I think is very similar to these calculations but isn't cited here. Might be worth looking into including this.

  2. Small point but on pg30 you say "some dielectric rods" but ADMX uses both dielectric and metallic rods depending on which iteration you are referencing (the 2009 paper you reference [108] I think used two copper plated rods even) . While their older paper I think used a combo of the two (arXiv:astro-ph/0603108)

  3. On pg30 (footnote 11). These results are live now (arXiv:2110.06096) maybe worth adding the citation.

  4. In Section 7.1 the parameters cited for each experiment are a little inconsistent. Most notably you cite the system temperature of 200mK for QUAX but no other experiment and the 8T B-field of ADMX but not for any other.

At a minimum I would suggest being more consistent (either include B/T for all or none). A better solution (although more time consuming) is that this section would benefit from a table with the key parameters (Q,B,T, mass range) from Eq.36 spelled out.

  1. In your discussion in 7.1 about going higher in frequency you might want to expand briefly on the idea of single-photon counting (mentioned briefly when citing [107]) as a means to combat smaller volume. Specifically you could mention the CARRACK experiment (arXiv:hep-ph/0101200) using Rydberg atoms to detect single photons. Or the more recent qubit search shown in (https://arxiv.org/abs/2008.12231).

These works aren't as mature so I understand not including them. But since you comment on some future work for increasing Q and V I think it could be worth adding a sentence or two on the above as they are promising paths forward (although I will admit some bias in recommending you add this).

  1. Typos/Grammar/Minor Comments

pg2. "SN" -> "SM"

pg3. "is also included" -> "are also included" pg3. "seeking for more in-depth treatment of some to the topics" -> "seeking a more in-depth treatment of some of the topics" pg3. "here presented" -> "presented here" pg3. "as an strategy" -> "as a strategy" pg3. "pNB" -> "pNG" pg3. "weak couplings coming from a a spontaneously" -> drop an "a" pg3. "less model constraints that" - >"less model constraints than"

pg4. "Viewed the QCD" -> "Viewing the QCD" pg4. "plays now" -> "now plays" pg4. "in absence" -> "in the absence"

pg5. "10^7 GeV^-1" -> I think it should be GeV right? pg5. "being g" -> "g being" pg5. "where g_ag" -> I think this should be "g_Ag" at this stage right? Or if not I think it needs to change in equation (6) so that it matches the description.

pg6. "quarks up and down" -> "up and down quarks" pg6. In the text you flip back to "g_ag" instead of "g_Ag" used elsewhere in this section pg6. "remaining of the report" -> "rest of the report" pg6. "much higher that the electroweak" -> "than" not "that" pg6. "are that do not contain" -> "are that they do not contain"

pg7. "issue that will be commented later" -> "an issue that will be commented on later"

pg8. "therefore are relevant to use astrophysics to contraint" -> "therefore it is relevant to use astrophysics to constrain" pg8. "very small coupling that arise by" -> "very small couplings that arise by" or "very small coupling that arises from"

pg9. "how are they constrained" -> "how they are constrained" pg9. "therefore be used to constraint" -> "therefore be used to constrain"

pg10. "Universe temperature" -> "Universe's temperature" pg10. "make the potential to "tilt"" -> "make the potential "tilt"" pg10. "wherever was down" -> "wherever it was down"

pg11. "is each scenario" -> "in each scenario" pg11. "an harmonic" -> "a harmonic" pg11. "for larger values corrections" -> "for larger values the corrections" pg11. (footnote) "vaccum" -> "vacuum" pg11. (footnote) "called sometimes" -> "sometimes called"

pg12. "the higher is its relic density" -> "the higher its relic density" pg12. "leading to an homogeneous" -> "leading to a homogeneous" pg12. "value of theta is just unknown" -> "value of theta is unknown"

pg13. "much lower that the above" -> "much lower than the above" pg13. "one assumes a theta is also" -> "one assumes theta is also"

pg14. "wall and strings" -> "walls and strings" pg14. "not a exotic" -> "not an exotic" pg14. "topological defects forms" -> "topological defects form" pg14. "vaccua" - > "vacua"

pg15. "vaccua" -> "vacua" pg15. "vaccum" -> "vacuum" pg15. "correspond to a wider" -> "corresponding to a wider"

pg16. "rest of species" -> "rest of the species" pg16. "put a upper" -> "put an upper" pg16. "expansion rate at inflation" -> "expansion rate of inflation" pg16. "longer that the age" -> "longer than the age"

pg17. "of the alter the" -> "or alter the" ?? pg17. "of in Figure 4" -> "in Figure 4" pg17. "field the drove inflation" -> "field that drove inflation" pg17. "Earch" -> "Earth"? pg17. "as byproduct of" -> "as a byproduct of"

pg18. "Each stage has also" -> "Each stage also has" pg18. "depends on their initial mass" -> "depends on its initial mass" pg18. "allows to reconstruct the evolutionary" -> "allows for the reconstruction of the evolutionarily" pg18. "case of the Sun" -> "case for the Sun" pg18. "a exotic cooling processes" -> "an exotic cooling process" or "exotic cooling processes"

pg19 (Fig2) "present is practically" -> "present in practically" pg19. "a exotic cooling" -> "an exotic cooling" pg19. "reconciling observations" -> "reconcile observations"

pg20. "axio-Bremstrahlung" -> "axio-Bremsstrahlung" pg20. "brehmstrahlung" - "bremsstrahlung" pg20. "WD are relatively" -> "WDs are relatively" pg20. "The evolution of the star follows then a simple" -> "Then the evolution of the star follows a simple" pg20. "better distance determinations data quality" -> This phrasing is confusing to me and I'm not sure what it means exactly?

pg21. "(see later)" -> Why not say see Section X.X? pg21. "strongest constrain" -> "strongest constraint" pg21. "a axion-nucleon interaction" -> "an axion-nucleon interaction" pg21. "considerable uncertainty remain" -> "considerable uncertainty remains" pg21. "but however it was later" -> either "but it was later" or "however it was later" pg21. "the constrains from NS" -> "the constraints from NS" pg21. "an homogeneous" -> "a homogeneous"

pg22. "field extend over" -> "field extends over" pg22. "axion mass is low enough" -> can you give an Order of Magnitude here? pg22. "Earch" -> "Earth" pg22. "compabitible" -> "compatible" pg22. "works have produced excluded region" -> "works have produced excluded regions" pg22. "there still is hint" -> "there still is a hint" pg22. "this observations" -> "these observations" or "this observation" pg22. "above described" -> "described above"

pg23. "observation of such gamma-ray" -> "observation of such a gamma-ray" pg23. "used to constraint" -> "used to constrain" pg23. "remaining of this text" -> "remainder of this text"

pg24. "assumption of the 100% of the dark matter" -> "assumption that 100% of the dark matter" pg24. "Another techniques in this category are" -> "Another techniques in this category is"

pg25. "an homogenous" -> "a homogenous" pg25. "being omega the energy" -> "and omega being"

pg26. "number of LSW experiments has been" -> "number of LSW experiments have been" pg26. "both making use" -> "both make use"

pg27. "UHE" -> not sure you define this acronym yet so should spell out "Ultra High Energy"

pg28. "larger that the" -> "larger than the"

pg30. "poses a experimental" -> "poses an experimental" pg30. "such signal" -> "such a signal" pg30. "proportional the time" -> "proportional to the time" pg30. "vaccuum" -> vacuum pg30. I would cite something for this SQL (probably the Haus-Caves theorem). Since there are many SQLs > C. M. Caves. Quantum limits on noise in linear ampli?ers. Phys. Rev. D, 26:1817{1839, Oct 1982. > H. A. Haus and J. A. Mullen. Quantum noise in linear ampli?ers. Phys. Rev., 128:2407{2413, Dec 1962 pg30. "includes additional component" - > "includes an additional component" or "includes additional components" pg30. "who has pioneered" -> "who have pioneered"

pg31. "compensated enhancing" -> "compensated by enhancing" pg31. "in the the future" -> "in the future"

pg32. "in this type of searches" -> "in these types of searches" or "in this type of search" pg32. "proving proven an improvement" -> not sure what this is supposed to say, maybe should be "giving an improvement"

pg33. "dynamics than the axion" -> "dynamics to the axion" pg33. "along this directions" -> "along this direction" pg33. "A conceptually similar is done" -> Missing a noun. Maybe you meant "A conceptually similar search is done" pg33. "split he ground" -> "split the ground" pg33. "in the in the" -> "in the"

pg34. "by a virtually any" -> "by virtually any"

pg35. "abundant that photons" -> "abundant than photons" pg35. "Despite of the above" -> "Despite the above"

pg35-36. The paragraph ending 35 and continuing to 36 is almost verbatim from the 1st paragraph of Section 4.1.2.

pg37. "similar that the" -> "similar to the"

pg38. "like a collections of " -> "like a collection of"

---

## Round 2 · Author Response

All the referees minor requests have been adopted

---

## Editorial Decision

published